# Temperature-triggered in situ forming lipid mesophase gel for local treatment of ulcerative colitis

Marianna Carone[1,5], Marianne R. Spalinger [2,5], Robert A. Gaultney[3,5], Raffaele Mezzenga [4], Kristýna Hlavačková[3], Aart Mookhoek[3], Philippe Krebs [3,6] ✉, Gerhard Rogler [2,6] ✉, Paola Luciani [1,6] ✉ & Simone Aleandri [1,6] ✉

Ulcerative colitis is a chronic inflammatory bowel disease that strongly affects patient quality of life. Side effects of current therapies necessitate new treatment strategies that maximise the drug concentration at the site of inflammation, while minimizing systemic exposure. Capitalizing on the biocompatible and biodegradable structure of lipid mesophases, we present a temperature-triggered in situ forming lipid gel for topical treatment of colitis. We show that the gel is versatile and can host and release drugs of different polarities, including tofacitinib and tacrolimus, in a sustained manner. Further, we demonstrate its adherence to the colonic wall for at least 6 h, thus preventing leakage and improving drug bioavailability. Importantly, we find that loading known colitis treatment drugs into the temperature-triggered gel improves animal health in two mouse models of acute colitis. Overall, our temperature-triggered gel may prove beneficial in ameliorating colitis and decreasing adverse effects associated with systemic application of immuno-suppressive treatments.

Ulcerative colitis (UC) is a chronic remitting-relapsing inflammatory disorder of the large intestine, involving the colonic and rectal mucosa[1]. Clinically, 75% of patients suffer from left-sided colitis or proctitis but the inflammation can spread upward in a continuous manner and involves the colon partially or entirely[2]. There is no known cure for UC and the chronic relapse and remission often result in patient disability[1,2]. All treatments currently recommended by the European Crohn's and Colitis Organization (ECCO) and American Gastroenterological Association (AGA) struggle to deliver the desired remission rates, and many patients must cycle through several different therapies to achieve remission[3,4]. Following a step-up approach,

the first-line treatment of mild to moderate left-sided UC or pancolitis is 5-aminosalicylic acid (5-ASA, combined topical and oral administration) for the induction of remission. For refractory patients, and, in severe disease cases, systemic corticosteroids, azathioprine, 6-mercaptopurine, monoclonal antibodies (such as infliximab, an anti TNF-α; vedolizumab, an anti $\alpha_4\beta_7$ integrin; and ustekinumab, IL-12/IL-23 blockade) and ozanimod (a sphingosine 1-phosphate receptor modulator) are the treatments of choice to obtain remission[4–8]. Biological-based therapies may have considerable side effects including systemic toxicity, resulting in recurrence of opportunistic infections, psoriasis, a lupus-like syndrome, and loss of response to therapy over time[9–13].

[1]Department of Chemistry, Biochemistry and Pharmaceutical Sciences, University of Bern, Bern, Switzerland. [2]Department of Gastroenterology and Hepatology, University Hospital Zurich, University of Zurich, Zurich, Switzerland. [3]Institute of Tissue Medicine and Pathology, University of Bern, Bern, Switzerland. [4]Laboratory of Food & Soft Materials, Institute of Food, Nutrition and Health, IFNH; Department for Health Sciences and Technology, D-HEST, ETH Zurich, Zurich, Switzerland. [5]These authors contributed equally: Marianna Carone, Marianne R. Spalinger, Robert A. Gaultney. [6]These authors jointly supervised this work: Philippe Krebs, Gerhard Rogler, Paola Luciani, Simone Aleandri. ✉e-mail: philippe.krebs@unibe.ch; gerhard.rogler@usz.ch; paola.luciani@unibe.ch; simone.aleandri@unibe.ch

Recently, tofacitinib (TOFA), a small-molecule inhibitor of the enzymes Janus kinase 1 and 3 (JAK1 and JAK3, respectively)[14], was approved by European and US regulators for the oral treatment of UC in patients who had intolerance or a loss of response to biologic drugs. Its oral administration is preferred by many patients over biologics in maintenance of remission and endoscopic improvement[15,16]. In steroid-refractory UC, the use of tacrolimus (TAC)—a macrolide that inhibits T-lymphocyte activation—is recommended[17]. TOFA and TAC, however, showed dose-dependent adverse effects when administered systemically (e.g., nephrotoxicity, thromboembolic complications, headache, metabolic disorders) in a significant fraction of patients[18–21], which may require discontinuation of the treatment in some cases and limits the dosages that can be administrated[21]. Taken together, the side effects of these systemically administered drugs must be weighed in patient management against the potential benefits of UC treatment. If not for the limiting side effects, higher drug concentrations would likely have a higher efficacy. The specific localization of the disease to the colon encourages the use of topical therapies[22]. Indeed, delivery via the rectal route is a safe therapeutic approach that can maximise the drug concentration directly at the site of inflammation while minimising systemic exposure. 5-Aminosalicylic acid (5-ASA) or budesonide in rectal preparations as enema or foam are routinely used as first-line treatment for UC[23,24]. The rectal administration of 5-ASA in UC patients has been shown to be significantly more efficient than oral administration[25–30]. The efficacy of rectal administration is further supported by the finding that steroid-refractory ulcerative proctitis has been managed by topically administered TAC as ointment[27,31], suppository[32] or enema[20,21,30]. Although clinical studies have shown that rectal 5-ASA preparations are more effective than oral preparations, these treatments are still rarely prescribed[33]. The efficacy of conventional enema-based formulations is intrinsically limited by their insufficient retention in the colon[34] and faecal urgency associated with the large volumes administered[35]. The required retention time—at least 20 min—together with frequent dosing negatively affect patient compliance[36]. One potential solution is a lipidic mesophase (LMP) based formulation, a versatile delivery system able to protect and release the incorporated drugs slowly in vivo[37–39]. Upon hydration, monoacylglycerol lipids such as monolinolein (MLO, generally recognized as safe for human and/or animal use by the FDA) can self-assemble in different arrangements. By increasing the water content, the less viscous lamellar (L) phase transforms first to an Ia3d and then to a Pn3m cubic phase (Q) which are similar in appearance and rheology to a highly viscous cross-linked hydrogel[40]. To overcome the hurdle of administering a highly viscous gel, not only water[41], but also temperature, can be used as a trigger to tune the viscosity of the system. Increasing the system's temperature, indeed, induces a transition from the L phase to a Q phase (with Ia3d symmetry)[40,42]. Considering the peculiarity of the rectal milieu, characterized by a low volume and with a composition highly affected by age, biological sex and pathology[43], water is not the most suitable trigger for an in situ gelation. Thus, rectal temperature is the ideal condition to activate the transformation of the precursor L phase into the cubic phase gel.

In this work, we present a gel platform that effectively employs rectal temperature as a trigger for the formation of a highly viscous adhesive depot system: a temperature-triggered in situ-forming lipid gel (TIF-Gel). This TIF-Gel (loaded with the hydrophilic TOFA or hydrophobic TAC) further enhances drug effectiveness in two established models of inflammatory bowel disease (IBD)[44,45] and is well retained after administration. Additionally, our data indicate that UC drug activity is better localized by delivery via the TIF-Gel. Overall, we expect that our TIF-Gel may translate to a topical mucosal therapy with high patient friendliness, concomitantly decreasing problems with retention, bloating, and urgency, and therefore increasing medical adherence.

## Results

### Physico-chemical characterization of TAC- and TOFA-loaded TIF-Gel

To decrease the drugs' adverse effects and to improve either the performance of the topical formulation or the therapeutic outcome of each drug, we designed and developed a gel formulation based on the concept that at 25 °C, and in the presence of a low percentage of water, MLO forms a lamellar (L) phase with a lower structural strength with respect to the cubic phase (Q), resulting in a formulation easier to administer and more able to treat remote tissue areas, as depicted in Fig. 1a. Once applied to the rectum, the precursor L phase gradually absorbs heat (and the available amount of water) from the body and rapidly (< 5 min) converts into the cubic phase, contributing to the formation of a depot in situ. The gel, therefore, allows local release of the incorporated drug in a sustained fashion. As a first step, we used small angle X-ray scattering (SAXS) measurements to determine the optimal amount of water needed to obtain a lamellar phase which provides a transition to the cubic phase at 38 °C. (Fig. 1b, c). The X-ray beam directed at the gel results in a scattering pattern with a set of maxima that correspond to sharp Bragg reflections characteristic of the long-range positional order. The sequence of Bragg reflections (and their ratio; listed in Fig. 1a) identifies the symmetry of the mesophase studied[46].

As shown in Fig. 1b, with 12% water the Bragg reflections characteristic of L phase were present at 25 and 38 °C. Hydrating the MLO to 14% water led to a lamellar structure at 25 °C and a coexistence of L and Q structures (with an Ia3d geometry) at 38 °C, whereas increasing the amount of water up to 18% w/w induced the L→Q transition already at 30 °C. On the other hand, a mesophase composed by 16% w/w of water and 84% w/w of MLO gives Bragg reflections characteristic of the lamellar structure at 25 °C and a transition to a Q structure (with a Ia3d geometry) at 38 °C, i.e., the rectal temperature.

The reflections characteristic of this L phase (containing 16% water) adopt those characteristic of a Q phase after only 5 min of incubation at 38 °C (Fig. 1c), indicating a rapid conversion of the lamellar precursor into the Ia3d cubic structure[40], making it particularly suited for rectal administration.

The transition is reversible if the temperature is brought back to 25 °C (see Supplementary Fig. 1). While this information is not relevant for rectal applications per se, it is an important property for the storage conditions of TIF-Gel.

The diverse topologies of the mesophases were confirmed by the different viscoelastic regimes identified by rheological (frequency sweep) measurements. Specifically, the precursor L phase had a low structural strength, as indicated by the lower value of storage modulus and loss modulus (G' and G", respectively) with respect to the viscoelastic Q phase. This resulted in a less viscous pseudoplastic gel characterized by extensive energy dissipation mechanisms associated with the parallel slip of the lamellae. In simulated administration conditions, increasing the temperature and water availability resulted in swelling of the structure corresponding to a Q phase transition (where both G' and G" are higher than those obtained for L phase—Fig. 1d). Moreover, either the flow or the yield points (both representing the shear limit above which a material starts to behave like a fluid) better determine the differences between our low viscous lamellar precursor and the high viscoelastic cubic gel and they might identify a threshold above which a formulation is too elastic to be rectally applied (Fig. 1e). Since the sliding of a lamellae can occur along any possible direction, a low shear is required to be applied to this gel so that it starts behaving like a fluid and it can be forced to pass easily through a canula for enema, a syringe or a colon pipe. This translates into a low viscosity material with a low structural strength easier to administer compared to the fully hydrated cubic gel owing to its high flow and a yield point (see also Supplementary Fig. 2).

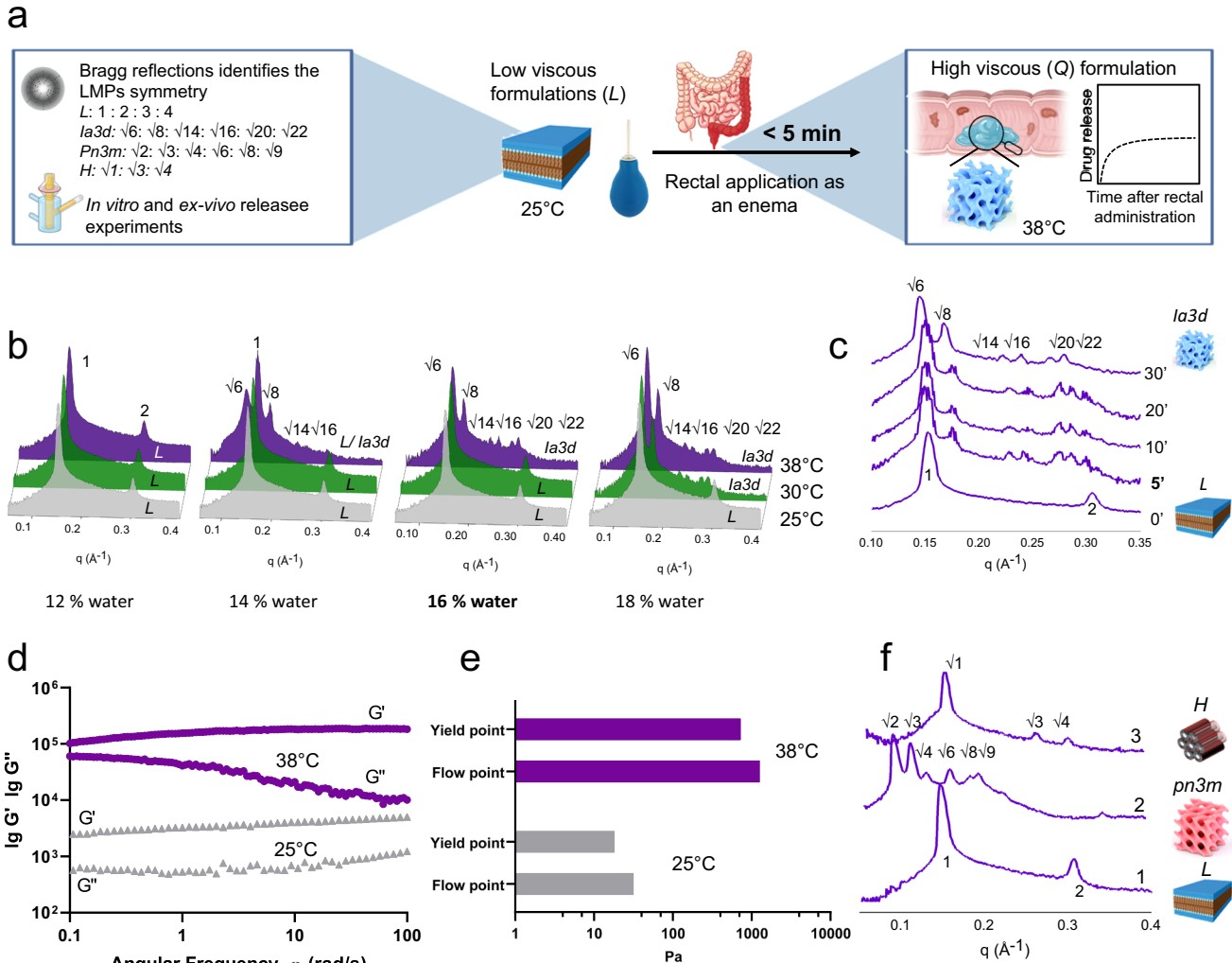

**Fig. 1 | In vitro characterization of the TIF-Gel. a** schematic depiction of the in vitro characterization and the mechanism of the gel formation. **b** SAXS spectra acquired at different temperatures (25, 30 and 38 °C; bottom, middle and top spectra, respectively) on gels containing increasing amounts of water (12%, 14%, 16% and 18 % w/w) and **c** SAXS spectra acquired at different times (5, 10, 20 and 30 min) after incubation at 38 °C; **d** frequency sweep at the end (purple symbols) and beginning (grey symbols) of the release experiments. **e** Flow and yield points obtained for the lamellar phase (grey bars) and cubic gel (purple bars). **f** SAXS before (1) and after incubation of LMPs in HEPES buffer (2) and in HEPES buffer enriched with 1000 U/mL of lipase (3). The LMP graphics (L; cubic ia3d, cubic pn3m and hexagonal) are adapted from ref. 38 with the permission of AIP (https://doi.org/10.1063/PT.3.4522) and from ref. 39 with permission of RSC (https://doi.org/10.1039/D2TB00403H). Additional graphics (Franz cell, gel and colon tract) were created using BioRender. Source data are provided as a source data file.

To prove the occurrence of the expected transition, a series of SAXS experiments was carried out after the gel was soaked in 1 mL of HEPES (or, alternatively, in a buffer solution containing lipase) and incubated at 38 °C for 8 h. As shown in Fig. 1f, the *L* phase absorbed heat and water during the release experiments reaching a cubic (*pn3m*) phase with a lattice parameters (a = 8.7 nm) and a water channel ($d_w$ = 4 nm) comparable with those obtained for a *Pn3m* phase at its maximum hydration level[47]. These transitions were also confirmed in vivo where, after rectal application, the gel excreted and collected with the stool after 30 min had an *Ia3d* phase identity, whereas the residual gel present in the colon after 6 h was determined to be in the *pn3m* cubic phase (see Supplementary Fig. 3).

The presence of lipase (100 U/mL) hydrolysed the ester group of MLO inducing a transition from *Q*→Hexagonal phase (*H*)[48], with the latter not linked to a burst release phenomenon (Fig. 2). Based on this initial characterisation, we chose an 84% MLO and 16% water formulation for subsequent in vitro and in vivo studies, which had suitable rheological properties to pass through a small diameter animal feeding needle (size 20 G) to further expand into a sponge-like system once injected into the rectum.

## Drugs are efficiently encapsulated and released from TIF-Gel

In order to properly use TIF-Gel as a treatment option, the thermal characteristics of the LMP should not be perturbed by the guest drug. To evaluate the influence of the active principles on the phase identity, TOFA (a hydrophilic inhibitor of the JAK1 and 3) and TAC (a hydrophobic immunosuppressive drug) loaded-mesophases were independently prepared and analysed with SAXS. Notably, the entrapment of drugs (5 mg of TOFA or 1 mg of TAC in 100 mg of gel−5 or 1% w/w, respectively) (Fig. 2a, b) did not affect the phase identity and thermal behaviour of the carrier gel and the rectal temperature still induced a transition from L→Q phase. When hydrated with water, the lipid/drug mixtures form the lamellar structure and the totality of the drugs are embedded in the gel with a 100% encapsulation efficacy. Moreover, both drugs do not form crystals once embedded in the 3D gel structure, as proven by the absence of reflections associated with a drug crystallization in the wide-angle X-ray scattering (WAXS) spectra. The drugs were also homogeneously distributed in the gel matrix (see Supplementary Fig. 4).

In the in vitro release experiments, drug-loaded TIF-Gel formulations were placed in the donor chamber of a vertical Franz cell (a

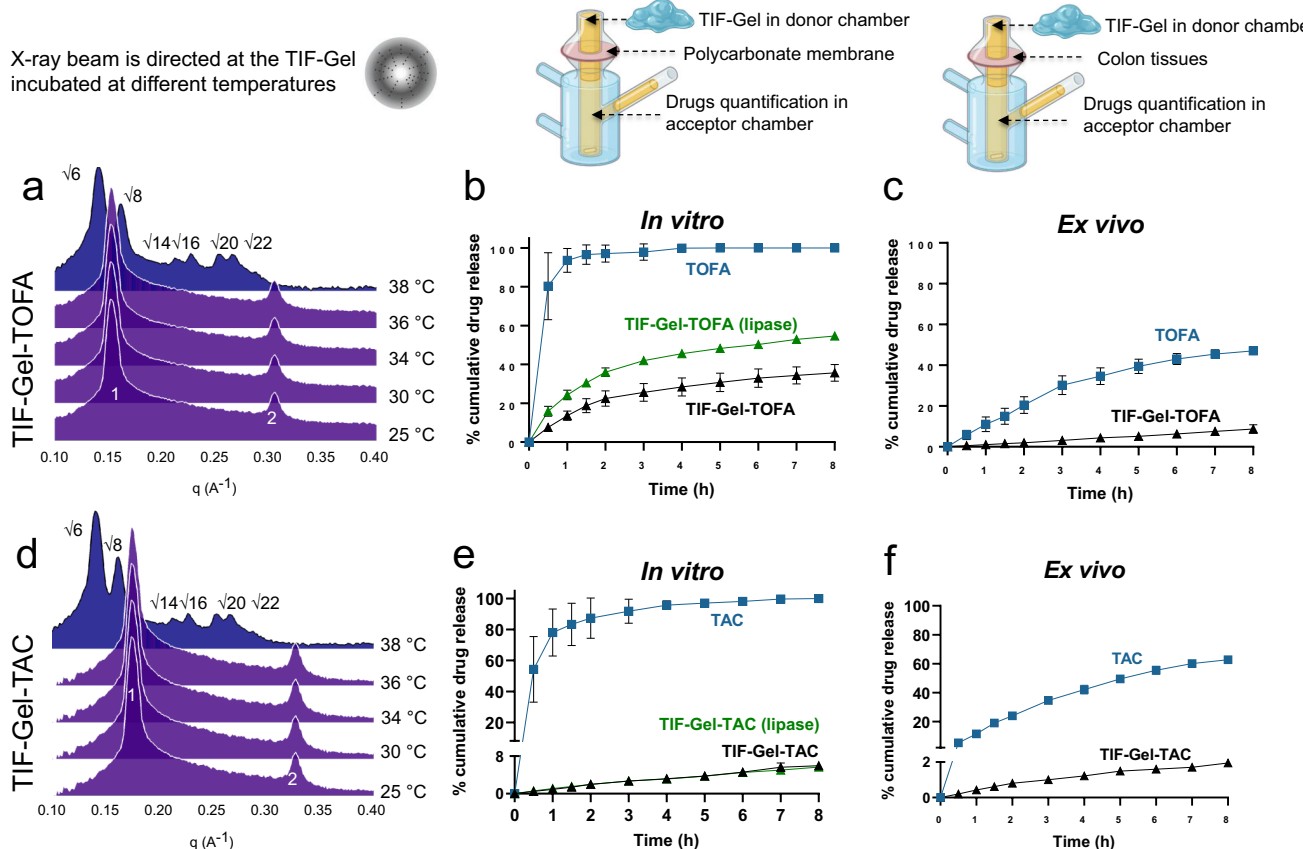

**Fig. 2 | In vitro and ex vivo characterizations of the drug-loaded TIF-Gel.**
**a** TOFA-loaded gel (TIF-Gel-TOFA) SAXS spectra acquired at different temperatures; **b** in vitro release of free drug (TOFA, blue line) and TIF-Gel-TOFA in HEPES buffer (black line) and in the presence of lipase (green line); **c** ex vivo release of free drug (TOFA, blue line) and TIF-Gel-TOFA (black line); **d** TAC loaded LMPs (TIF-Gel-TAC) SAXS spectra acquired at different temperatures; **e** in vitro release of free drug (TAC, blue line), TIF-Gel-TAC (black line) and in the presence of lipase (green line); **f** ex vivo release of free drug (TAC, blue line) and TIF-Gel-TAC in HEPES buffer (black line). Results in (**b**, **c**, **e**, **f**) are reported as mean ± STDV ($n = 3$). Cartoons (Franz cell, gel and X-ray pattern) are created by BioRender. Source data are provided as a source data file.

commonly used apparatus to assess the drug release from a semisolid dosage formulation in pre-clinical studies; depicted in Fig. 2) and kept separated from the acceptor chamber by a polycarbonate membrane with pore diameter of 3 µm, which allows the passage of the free drugs only[49]. Contrary to the small intestine, for which different in vitro models are established[50–52], for colon tissues only animal models are available and are mostly used in pre-clinical studies[53,54]. To bypass this limitation, we employed an ex vivo approach in which tissues isolated from healthy rat colon were used as natural membrane[55], replacing the polycarbonate membrane detailed above. The 3D gel network retains the TOFA (hydrophilic drug) and slowly releases it in both the in vitro and an ex vivo setups (Fig. 2b, c, respectively). The same sets of experiments were also carried out for the TAC-loaded gel. Results for this hydrophobic drug mirrored those of the hydrophilic TOFA in both in vivo and ex vivo experiments, reflecting the flexibility of this vehicle (Fig. 2e, f). Notably, the presence of lipase in our experimental conditions did not induce disassembly of the gel with consequent burst release of the drug, as reported for another lipid-based gel, developed to topically treat UC[56]. In this study, addition of the enzyme (*Thermomyces lanuginosus* lipase) induced a responsive release (+20% of drug released) from the hydrogel only after 24 h. In comparison, TAC and TOFA were released from our TIF-Gel within only 8 h, a time span more compatible with the retention time of rectally administered dosage forms. In 2015, Martiel et al. developed the structural control efficiency index (SCEI)[57], which provides an estimate of the kinetics of drug release for various phases. However, in our case, the phase identity of the gel changes dynamically during the release experiment. Thus, we

cannot use the above-mentioned paradigm to describe the release profile. Indeed, our hydrophobic drugs do not follow a Fickian diffusion profile and, consequently, the release profile cannot be modeled using the Higuchi equation. We did not observe any gel erosion (no weight loss was recorded either in vitro or in ex vivo experiments) and we can, therefore, reject the hypothesis that the release process is driven by gel dissolution. Interestingly, the presence of 10 mg of the single drugs (10% w/w of both TOFA and TAC) did not affect the phase identity and the transition temperature of the gel, which gives the desired lamellar phase at room temperature and the cubic (*Ia3d*) phase at 38 °C (see Supplementary Fig. 5). This demonstrates that the administration of a low volume of TIF-Gel could indeed deliver high amounts of drugs, likely reducing the urgency associated with the intra-rectal application of large volumes[58].

### Effect of TIF-Gel-TOFA on dextran sulfate sodium (DSS)-induced acute colitis
To test the potential efficacy of the gel in treating an acute UC flare-up, we applied TIF-Gel loaded with TOFA to a mouse model of acute colitis induced by dextran sulfate sodium (DSS). DSS is toxic to epithelial cells and its application compromises the integrity of the intestinal barrier, thereby leading to an erosion of the epithelium and activation of submucosal immune cells by intestinal microbes[59]. Through experimentation, we determined that application of the gel every other day yielded robust mitigation of local and systemic inflammation.

Mice treated with this regimen of TIF-Gel-TOFA displayed decreased weight loss and disease severity when compared to mice

treated with an empty TIF-Gel (Fig. 3a, b). In contrast, drug in vehicle solution (TOFA), while improving weight loss, did not improve the disease score in these mice (Fig. 3b). Of note, daily application of the compounds did not yield as robust results, and the differences between free TOFA and TIF-Gel-TOFA were less apparent under this regimen (see Supplementary Fig. 6). Signs of systemic inflammation, determined by spleen size and cellularity, were also alleviated in mice treated every other day with the TIF-Gel-TOFA (Fig. 3c, d). Furthermore, local pro-inflammatory cytokine levels were reduced in TOFA and TIF-Gel-TOFA-treated mice, and anti-inflammatory IL-10 levels were increased only in the TIF-Gel-TOFA group (Fig. 3e). Local inflammation was also mitigated by the TIF-Gel-TOFA as determined by a reduction in colon shortening and pathology (Fig. 3f–h). For colon shortening, but not for histopathology, TIF-Gel-TOFA was more effective than the drug in vehicle, and no differences were detectable in the proportion of immune cell populations of the spleens or mesenteric lymph nodes of the different treatment groups (see Supplementary Fig. 7). Overall, these data indicate that a topically applied temperature-dependent in situ-forming gel carrying TOFA represents a valuable tool to mitigate acute intestinal inflammation.

### Effect of TIF-Gel-TAC on T-cell transfer colitis

TIF-Gel acts as a platform able to host and release molecules with different polarities (see Fig. 2). Thus, we also assessed the ability of the TIF-Gel loaded with the hydrophobic drug TAC to reduce colitis severity using a model of T cell-mediated colitis, namely the T cell transfer colitis model[60]. In this model, naive CD4 + T cells are transferred into B and T cell-deficient $Rag^{-/-}$ recipient mice, which results in the development of T helper cells that react against luminal antigens and subsequently induce a strong colon inflammation[61–63]. Three days after naive T cell transfer into $Rag^{-/-}$ hosts, mice were treated with 100 µL (i) TAC-loaded TIF-Gel (TIF-Gel-TAC), (ii) empty TIF-Gel (TIF-Gel) or (iii) TAC in vehicle solution (TAC) via daily rectal instillation (Fig. 4a). Weight development and monitoring of disease activity demonstrated that mice that received empty TIF-Gel or TAC in vehicle solution started to develop the first signs of colitis around day 10 post T cell transfer as evidenced by progressive weight loss and signs of diarrhoea (Fig. 4b, c). Of note, mice that were treated with TIF-Gel-TAC did not lose weight and diarrhoea scores were lower than in the other two groups (Fig. 4b, c). Moreover, all mice receiving TAC (either in TIF-Gels or administered in vehicle) showed longer colons and reduced spleen weight (Fig. 4d), indicating reduced disease in these two groups when compared to mice treated with empty TIF-Gels.

On day 19 after T cell transfer, all mice were subjected to colonoscopy to evaluate macroscopic signs of colitis. Interestingly, TAC-administration via TIF-Gels as well as TAC administration in vehicle reduced endoscopic signs of colitis. Although there was a clear trend towards further reduction of endoscopic scores in mice that received TIF-Gel-TAC, this was not significant (Fig. 4e). In contrast, and in line with disease activity scores, mice that received TIF-Gel-TAC did not only show clearly reduced colitis severity when compared to mice that were treated with empty TIF-Gels, but also when compared to mice that received TAC in vehicle solution (see histology score; Fig. 4e). Taken together, these data clearly indicate that TAC administration via TIF-Gels is superior in reducing colitis severity than TAC-administration in vehicle.

T cell transfer colitis is mainly mediated by aberrantly activated T helper cells, and especially IFN-γ+ (Th1) and IL-17+ (Th17) CD4 + T cells contribute to the disease. To test the effect of TAC administration either in vehicle or in the TIF-Gels, we analysed proportions of T helper cells in the colonic lamina propria (Fig. 5a), mesenteric lymph nodes (Fig. 5b) and the spleen (Fig. 5c). Of note, both TAC administration forms reduced the relative abundance of T cells in the *lamina propria*, mesenteric lymph nodes and the spleen (Fig. 5a, c). Among those, Th1 and Th17 cells were reduced with TAC in vehicle as well as with TIF-Gel-

TAG when compared to the mice that received empty TIF-Gels only (Fig. 5). While there was no difference among Th1 cells between mice receiving of TAC in vehicle and those receiving TIF-Gel-TAC, the reduction in Th17 cells was significantly more pronounced in mice receiving TIF-Gel-TAC than in those receiving TAC in vehicle (Fig. 5). In general, there was not much effect on the abundance of FoxP3+ (regulatory) T cells (Fig. 5). These findings were also reflected in cytokine measurements in colonic tissues (Fig. 5d), where we found reduced levels of IFN-γ and IL-17 in mice treated with the free drug. TIF-Gel-TAC further reduced levels of these two cytokines and in addition also significantly reduced levels of TNF-α (Fig. 5d), indicating that TIF-Gel-TAC was more effective at reducing production of pro-inflammatory cytokines than free drug alone. In summary, these results indicate that administration of TIF-Gel-TAC is superior in reducing disease-promoting T helper cells in the setting of T cell induced colitis.

### Rectal drug delivery via TIF-Gel reduces systemic drug exposure

To demonstrate that rectal TIF-Gel application was indeed suitable for minimizing systemic drug release, we analysed drug release in vivo by longitudinally monitoring drug plasma levels of mice after colonic TIF-Gel enema. For this purpose, healthy mice received a single enema of either drug-loaded TIF-Gel (TIF-Gel-TOFA or TIF-Gel-TAC) or of free drugs (TOFA or TAC) and plasma drug concentrations were measured at different time points (Fig. 6a). Mice receiving free TOFA had an early peak in plasma concentration at 0.25 h (Fig. 6b); TOFA plasma levels rapidly decreased thereafter, following first-order kinetics. In mice receiving TIF-Gel-TOFA, the peak concentration at 0.25 h was significantly lower. The area under the curve (AUC), a measurement of cumulative systemic drug absorption, was also significantly reduced in the mice treated with TIF-Gel-TOFA when compared to the group treated with free TOFA (Fig. 6d). Administration of TAC, either as free drug, or as drug-loaded gel resulted in a low (and negligible) systemic drug circulation (Fig. 6c) and no difference was detected in their AUCs (Fig. 6e).

## Discussion

The management of UC does not simply rely on the choice of a timely pharmacological treatment since the potential benefits of parenteral drug administration must always be weighed against concomitant side effects due to systemic circulation of the drug[64]. Thus, the specific localization of the disease should indeed encourage the pursuit of a rectal administration through which the drug directly reaches the site of inflammation with minimal systemic exposure[36]. We demonstrated here that our TIF-Gel can host and release drugs of different polarities in a sustained manner, and that the rectal temperature can be employed as a stimulus to transform the plastic (and low viscosity) lamellar precursor into the structured (and high viscoelastic) cubic gel in situ, while no available enema or foam can control drug release.

Among the various acyl glycerol lipids (such as phytantriol and monoolein) capable of forming lipidic mesophases in water via self-assembly, MLO (affirmed as GRAS for human and animal use by the US Food and Drug Administration) is a widely used material for encapsulating a broad range of drugs with various sizes and polarities. Its phase diagrams are unique, as at any temperature and water content it does not present coexistence of mesophases[65]. Consequently, changing the temperature (without increasing the water content) results in a direct transformation from lamellar to Ia3d cubic structure.

Hence, at 25 °C and in the presence of 16% w/w percentage of water, MLO forms an L phase with a low structural strength resulting in a formulation that is easy to apply and able to reach more remote areas of the colon. On the other hand, the pseudoplastic precursor has a higher viscosity than commercially available enemas such as Asacol® and Pentasa® and foam-containing 5-ASA and budesonide. Thus, once applied, our TIF-Gel adheres to the colon wall and it is retained for at

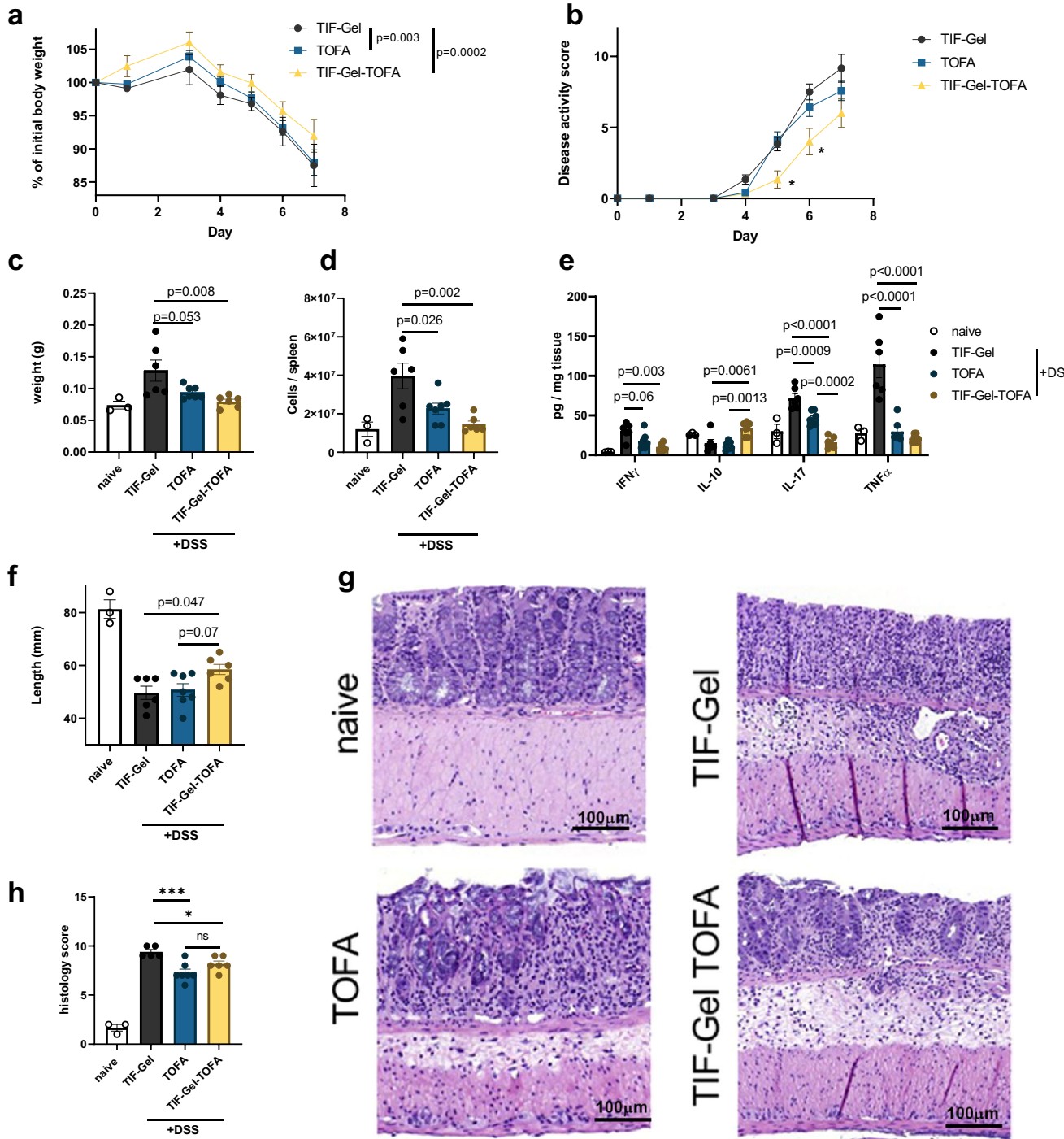

**Fig. 3 | TIF-Gel-TOFA effectively mitigates intestinal inflammation and disease induced by DSS treatment in mice.** Mice were prophylactically treated rectally with either empty gel (TIF-Gel; $n = 6$), tofacitinib in vehicle (TOFA; $n = 7$), or TOFA loaded-gel (TIF-Gel-TOFA; $n = 6$) and thereafter challenged with 2% DSS in the drinking water. Treatments were then applied every other day until the end of the experiment. Weights (**a**) and disease score (**b**) were recorded throughout the experiment. At the end of the experiment, spleens, mesenteric lymph nodes (mLNs) and colons were removed from the mice. The spleens were weighed (**c**) and single splenocytes were enumerated (**d**). The tissue concentrations of various cytokines were measured (**e**). The mouse colon length was measured (**f**), and the colon was opened transversally, cleaned, and prepared for histology (**g**). Colon histopathology scores were determined by a blinded pathologist and aggregated (**h**). *$P < 0.05$, **$P < 0.01$, ***$P < 0.001$, ****$P < 0.0001$, and actual value is provided for values less than 0.1 but not meeting significance threshold as determined by two-way ANOVA (**a**), multiple Student's *t* tests with Holm–Sidak correction for multiple comparisons (**b**), and one way ANOVA with multiple comparisons and Tukey correction (**c**, **d**, **e**, **f**, **h**). All tests were performed using Prism (GraphPad) and applying default settings for the above-mentioned analyses; naive values were excluded from analyses; all error bars are ± SEM. Source data are provided as a source data file.

least 6 h, a time needed to avoid loss of material[40,66,67] (see Supplementary Fig. 8). Moreover, once applied to the rectum, the precursor L phase gradually absorbs heat (and the available amount of water) from the body and converts into the cubic phase, which forms the sustained release depot in situ. In comparison to the available FluidCrystal technology® developed by the Swedish company Camurus[68,69], which consists of an alcoholic lipid solution that transforms to a gel upon contact with water, TIF-gel is not only dependent on water content, but

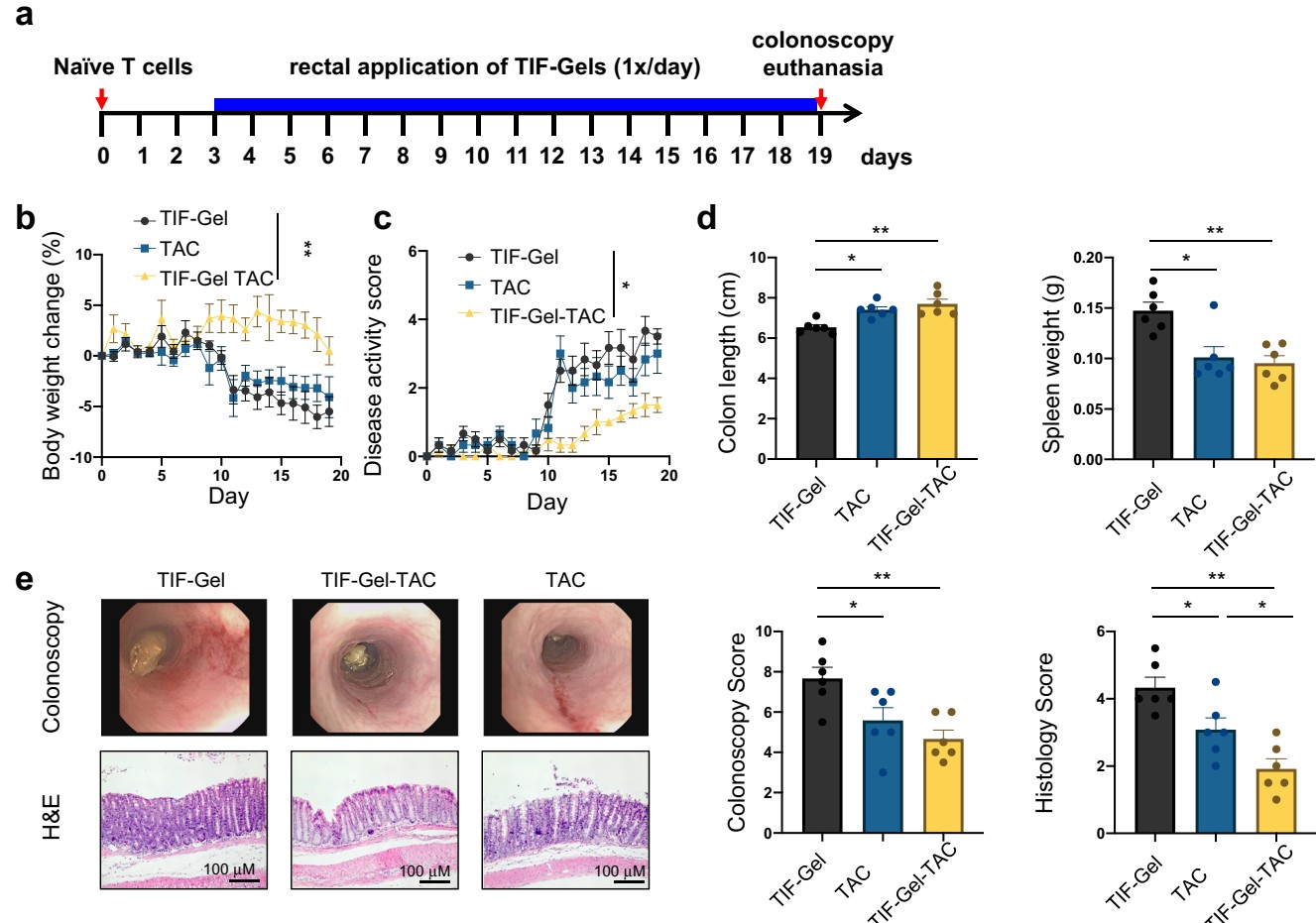

**Fig. 4 | Assessment of the effect of TAC-loaded TIF-Gel on T cell-mediated colitis.** 12–15-week-old *Rag*[-/-] mice develop colitis via transfer of $2.5 \times 10^5$ naive CD4 + T cells. Starting on day 3 after T cell transfer, mice (*n* = 6) received daily rectal instillations with TIF-Gel without drug (TIF-Gel), TAC-loaded TIF-Gels (TIF-Gel-TAC) or TAC in vehicle (TAC). **a** Schematic overview on the experimental set-up; **b** weight development over the course of the experiment; **c** cumulative disease activity score; **d** colon length and spleen weight; **e** representative pictures and respective scoring from mouse colonoscopy on day 19 post T cell transfer and from H&E-stained sections of the terminal colon collected on day 19 post T cell transfer. *$P$ < 0.05, **$P$ < 0.01 as determined by two-way ANOVA (**b**, **c**) and one way ANOVA with multiple comparisons and Tukey correction (**d**, **e**). All tests were performed using Prism (GraphPad) and applying default settings for the above-mentioned analyses; all error bars are ±SEM. Source data are provided as a source data file.

also uses temperature as a trigger for an in situ gelation. This aspect is of particular relevance since the volume of rectal fluid is low and highly affected by age, biological sex and pathology. In addition, TIF-gel is less fluid than a lipidic solution thereby avoiding a loss of material upon rectal application and, in contrast with liquid crystal technology, no initial burst release of drug was observed with our gel. Indeed, from our observations regarding drug retention, we are prone to believe that the release is regulated by the log P and molecular weight of the drug; and by the ability of the drug to pass across the lipidic layers and diffuse into the environment. The validation of the gel efficacy to treat colitis was assessed in two different animal models, namely DSS-induced colitis and the T cell transfer colitis model. The dual approach was crucial to ascertain that the ability of our gel to treat the diseases is independent from the specific colitis trigger. DSS works through damaging the epithelial layer of the intestine, allowing for unfettered microbial-immune cell interactions and thereby results in acute and severe colitis mimicking acute disease flares observed in human UC[70]. As topical TOFA has been applied to other inflammatory models[71], we postulated that this drug would be useful for testing the suitability of the TIF-Gel for application in the inflamed colon. Systemic and local inflammation were both reduced by the application of TOFA loaded into TIF-Gel when compared to control DSS-challenged mice treated with empty gel or TOFA in vehicle solution. Interestingly, daily rectal

administration of the drug in either form did not yield improved results and reduced the differences between the gel-loaded and free drug, suggesting an enhanced time-release advantage of the drug when delivered as TIF-Gel-TOFA. Histopathological scores between the TIF-Gel and vehicle TOFA were similar at endpoint, although by this time disease activity scores had normalized between the groups, and future studies with a less severe treatment regimen or altered time course (e.g., chronic low concentrations of DSS) might show more robust differences. The gel efficacy is further supported by the changes in cytokines measured from colon tissue samples, in which the most robust changes in these inflammatory mediators were observed in the TIF-Gel-TOFA treated mice. While DSS-induced colitis is mainly driven by the disruption of the intestinal epithelial barrier, the transfer colitis model is T cell-mediated; thus, this model represents a different aspect of human IBD. In T cell transfer-mediated colitis, local administration of TAC using the TIF-Gel was superior in reducing colitis severity when compared to application of TAC in vehicle solution. This further confirms the results in the DSS model and supports the observation that the TIF-Gel represents a valid option to locally deliver drugs to the inflamed mucosa. Compared with other depot systems, our gel presents several potential advantages. First, TIF-Gel is made from a nontoxic, GRAS compound and, differently from other in situ forming implants (made by triglycerol monostearate or

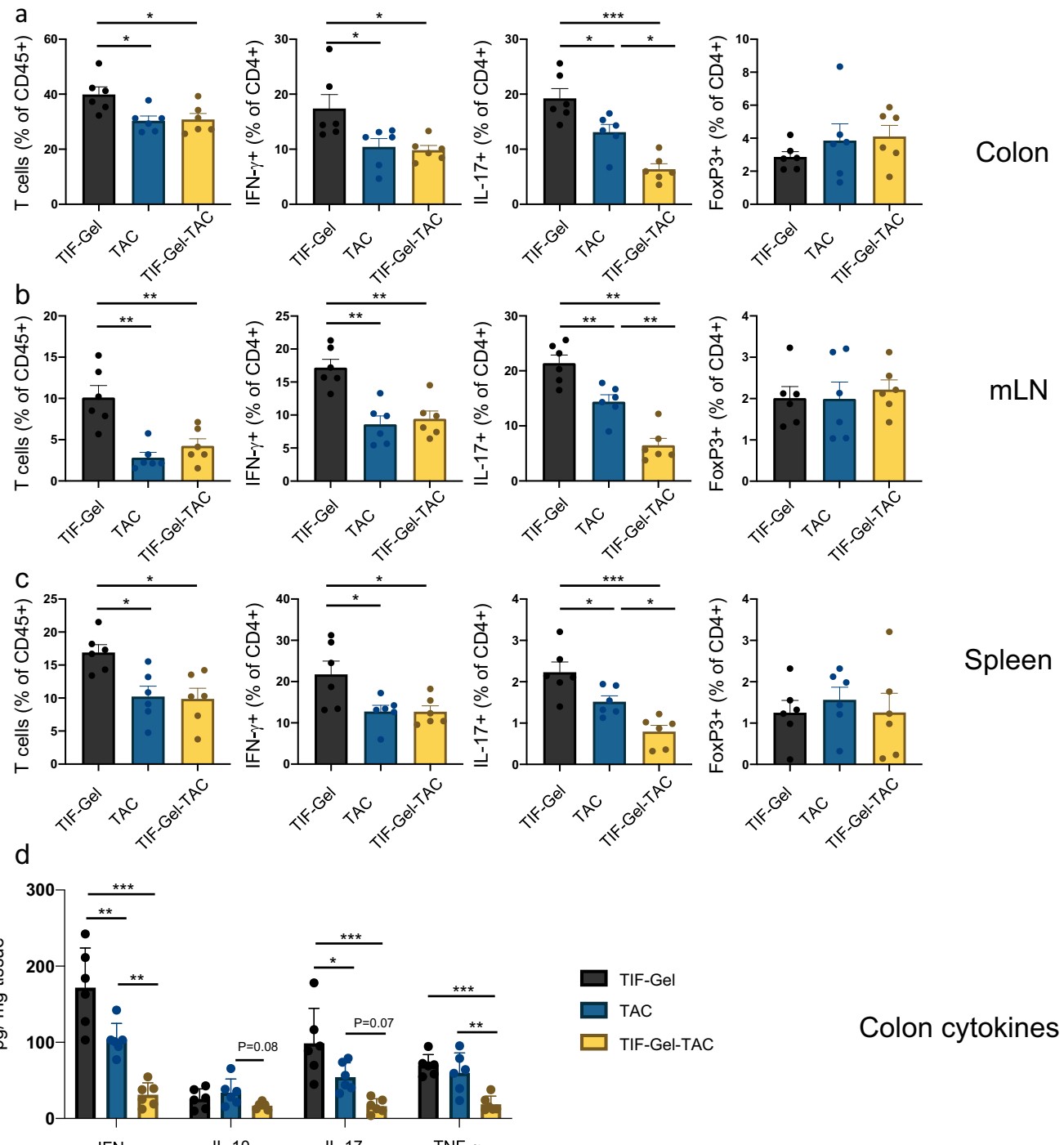

**Fig. 5 | Immune cell populations and cytokine levels in the colon from TAC-loaded TIF-Gel (TIF-Gel-TAC) treated mice.** 12–15-week-old *Rag*⁻/⁻ mice develop colitis via transfer of $2.5 \times 10^5$ naive T cells. Starting on day 3 post T cell transfer, mice received daily rectal instillations with TIF-Gel without drug (TIF-Gel), TAC-loaded TIF-Gels (TIF-Gel-TAC) or TAC in vehicle (TAC). Depicted are the relative abundance of the indicated cell populations in (**a**) the colonic *lamina propria*, **b** mesenteric lymph nodes; and **c** the spleen on day 19 after T cell transfer, and **d** levels of indicated cytokines in colon lysates. *$P < 0.05$, **$P < 0.01$, ***$P < 0.001$ as determined by one way ANOVA with multiple comparisons and Tukey correction. All tests were performed using Prism (GraphPad) and applying default settings for the above-mentioned analyses; all error bars are ±SEM. Source data are provided as a source data file.

poly(lactic-*co*-glycolic acid), PLGA) currently in clinical trials or commercially available, it does not require organic solvents (that may be toxic) during its in situ formation. Differently from what observed with other in situ forming gels, the L→Q transition is not associated with burst release of the drug. Noteworthily, MLO is relatively inexpensive and available in large quantities at a high grade of purity (Good Manufacturing Practice or Food Grade); the manufacturing is a single-step process, and both the tested drugs and the TIF-Gel exhibit long-term stability (see Supplementary Figs. 9 and 10). All these conditions facilitate a straightforward transfer of the technology from the bench to an industrial scale. Lastly, a low volume (0.1 mL) of TIF-Gel is able to deliver high amounts of drugs (up to 10 mg). Differently from the recently developed hydrogel for local drug delivery in IBD in which a low proportion of dexamethasone was encapsulated[72], our bulk gel is

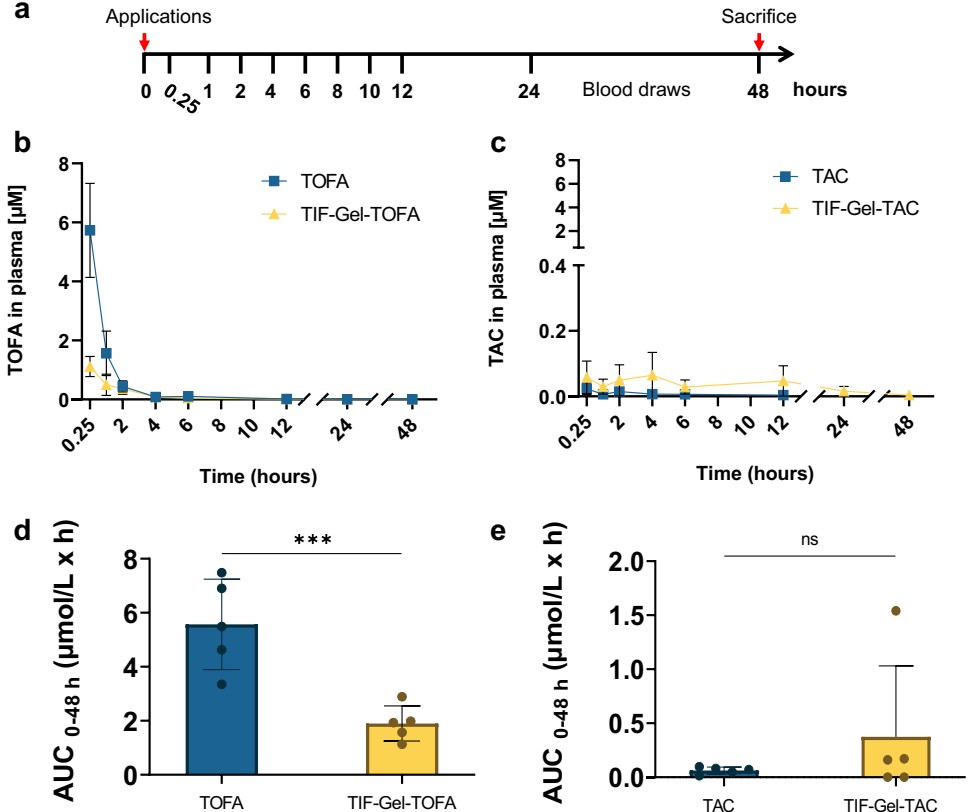

**Fig. 6 | Drug delivery via TIF-Gel leads to a low systemic drug exposure.**
**a** Experimental design for the pharmacokinetic study. Healthy mice ($n = 5$/group) received a single enema of either drug-loaded TIF-Gel (TIF-Gel-TOFA or TIF-Gel-TAC) or free drugs (TOFA or TAC). The plasma drug concentrations were measured at the indicated time points after administration. Plasma concentration versus time profiles of the pharmacokinetic experiment of TOFA- (**b**) and TAC-treated animals (**c**) and Area Under the Curve (AUC)$_{0-48h}$ values of TOFA- and TAC-treated mice (**d**, **e**, respectively). ***$P < 0.001$ as determined by two tailed Student's $t$ test. All tests were performed using Prism (GraphPad); all error bars are ±SD. Source data are provided as a source data file.

able to provide 100% drug loading (5 mg TOFA/100 mg gel and 1 mg TAC/100 mg gel) and still ensure a sustained release in vitro, ex vivo and in vivo. Moreover, 10% w/w of both TOFA and TAC can be loaded without affecting the phase identity and the transition temperature of the gel, which gives a lamellar phase at room temperature and a cubic (*Ia3d*) phase at 38 °C. These drug concentrations are higher than those contained in the commercially available enemas for UC treatment, which contain only 4% w/w of 5-ASA (in case of Asacol®) or 2% w/w of budesonide (for Budenofalk® and Entocort®). Although the drug concentrations in the TIF-Gel can reach high levels, this delivery platform also helps to simultaneously minimise systemic circulation. Indeed, as demonstrated by our pharmacokinetics study, the plasma concentration of the TOFA in TIF-Gel-treated mice was lower than in animals rectally treated with free drug within a 48 h period after application. On the other end, the pharmacokinetics of TAC either as free drug or as drug-loaded gel results in a negligible systemic drug absorption and no difference was detected in their AUCs. In a previous reported clinical study, rectal administration of TAC via a suppository resulted in a systemic exposure with a relative bioavailability of ~70% after 24 h in comparison to the oral formulation[73]. Here, upon administration, patients were asked to remain in a semi-recumbent position for 3 h (without defecation), a practice which facilitates drug absorption. Differently from this study, after administration our mice were left free to move and this introduced a certain external variability in the excretion time of the stools and/or of the free drug. The low residence time in the intestine could explain the low plasmatic levels of serum TAC observed in the free-drug treated mice, whereas the simultaneous low drug absorption of TAC released from TIF-Gel-treated mice could be explained by a slow and continuous drug release from the gel coupled with its hydrophobicity. In conclusion, this reduced systemic drug concentration, particularly in the TOFA-treated mice, indicates that the use of TIF-Gel could reduce the number and severity of systemic side effects resulting from the use of potent drugs such as TOFA and TAC in a clinical setting. Taken together, in addition to making the manufacture of this gel formulation very efficient, administration of low volumes to the rectum may help in improving the patients' adherence to the therapy, a goal not always achieved because of the fecal urgency experienced as a consequence of the high volumes of daily applied enemas. We envision that our formulation may reduce not only the drug side effects associated with systemic therapy but also the drawbacks related to the use of the rectal formulations commercially available.

In summary, our results demonstrate that TIF-Gel provides a valuable approach to effectively administer TOFA and TAC locally to the colonic mucosa resulting in sustained drug release. Our findings suggest that TIF-Gel enhances the localized activity of these anti-inflammatory drugs while potentially reducing the risk of side effects. We expect that TIF-Gel will broaden the portfolio of the currently available platforms for topical therapies owing to a higher patient friendliness and a reduced leakage, while concomitantly decreasing problems with retention, bloating and urgency thus increasing patient compliance to drug regimens.

## Methods
### Ethics statement
The work presented herein complies with all appropriate ethical regulations including animal welfare by the Cantonal Veterinary Offices of Berne and Zurich and in agreement with regulations of the Canadian Council on Animal Care (CCAC).

## Study design

The goal of this study was to design a drug delivery system (TIF-Gel) for UC. We hypothesized that temperature could be used as a trigger to induce a formation of a viscous gel in situ and it should rapidly adhere to inflamed rectum mucosa and release its cargo. Firstly, we characterized the TIF-Gel in vitro; subsequentially, we examined whether drugs delivery (TOFA and TAC) via TIF-Gel would affect therapeutic efficacy in two mouse models of UC, DSS-induced colitis and the mediated T cell model in accordance with institutional and federal regulations governing animal care and use, according to the ARRIVE guidelines. For experiments, 6 mice were used per group. All animals were included in the analysis. Histopathology was analysed by an experienced gastrointestinal pathologist blinded to group assignment using an established scoring system.

## Materials

Monolinolein (MLO) was purchased by NU-Check Prep, Inc. (MN, USA). Ultrapure water of resistivity 18.2 MΩ.cm was produced by Barnstead Smart2pure (Thermo Scientific) and used as the aqueous phase. Methanol, acetonitrile, and tetrahydrofuran were analytical grade supplied by Fisher Scientific (Schwerte, Germany). Ethanol absolute >99.5 wt% was obtained from VWR chemicals BDH (London, UK). Tofacitinib citrate (TOFA) was purchased by LC laboratories (Woburn, MA) and tacrolimus (TAC) was obtained from R&S Pharmchem Co., Ltd (Shangai, China). The lipase from porcine pancreas and methylcellulose (viscosity 25 cp) were obtained from Sigma Chemical Co. (St. Louis, USA). Caffeine (Ph. Eur. Quality) was purchased from Hanseler Swiss Pharma. HEPES salt was obtained from Carl Roth (Karlsruhe, Germany).

## Gel preparation

MLO was used as the lipid component of the mesophases and mixed with TOFA (5% w/w; 5 mg/100 mg) or TAC (1% w/w; 1 mg/100 mg). Lipid/drug mixtures were prepared by dissolving the appropriate amounts of lipid and drug stock solutions together in ethanol. The solvent was then completely removed under reduced pressure (freeze-drying for 24 h at 0.22 mbar) and the dried lipid mixture was hydrated by mixing weighed amounts of water in the presence or absence of 10% w/w TOFA or TAC in sealed Pyrex tubes and alternatively centrifuging (10 min, 5000×g) several times at room temperature until a homogenous mixture was obtained. The mesophase was then equilibrated for 48 h at room temperature in the dark. For in vivo studies, after 48 h equilibration (as described above) the formulation was loaded into a 1-mL syringe (Injekt-F, Braun) and the dead volume of the animal feeding needle (20 G, L × diam. 1.5 in. × 1.9 mm) for rectal administration was calculated so that exactly 100 mg was applied (see Supplementary Fig. 11). To check for drug homogeneity, gel loaded with TAC or TOFA was prepared as described above and transferred into a 2 mL Eppendorf tube. The tube was centrifuged (5 min, 5000×g) and kept at rest for 24 h. Subsequently, the gel was divided into three different layers (Top, Middle, and Bottom), and the drug content evaluated in each (see Supplementary Fig. 4).

## Small angle X-ray scattering

SAXS measurements were used to determine the phase identity and symmetry of the produced LMPs. Measurements were performed on a Bruker AXS Micro, with a microfocused X-ray source, operating at voltage and filament current of 50 kV and 1000 µA, respectively. The Cu Kα radiation (λCu Kα = 1.5418 Å) was collimated by a 2D Kratky collimator, and the data were collected by a 2D Pilatus 100 K detector. The scattering vector Q = (4π/λ) sin θ, with 2θ being the scattering angle, was calibrated using silver behenate. Data were collected and azimuthally averaged using the Saxsgui software to yield 1D intensity vs. scattering vector Q, with a Q range from 0.001 to 0.5 Å⁻¹. For all measurements, the samples were placed inside a stainless-steel cell between two thin replaceable mica sheets and sealed by an O-ring, with a sample volume of 10 µL and a thickness of ~1 mm. Measurements were performed at 25, 30, 34, 36 and 38 °C. Samples were equilibrated for 10 min before measurements, whereas scattered intensity was collected over 30 min and over 60 min in case of lamellar phase. On the other hand, for the kinetic study, the sample was pre equilibrated at 25 °C and inserted in the sample holder kept at 38 °C and the scattered intensity collected over 5 min. To determine the structural parameters such as the size of the water channels, SAXS data information on the lattice were combined with the composition of the samples[40].

## Rheology experiments

A stress-controlled rheometer (Modular Compact Rheometer MCR 72 from Anton Paar, Graz, Austria) was used in cone-plate geometry, 0.993° angle, and 49.942 mm diameter. The temperature control was set either at 25 or 38 °C. First, a strain sweep was performed at 1 Hz between 0.002 and 100% strain to determine the linear viscoelastic regime (LVR), the yield and flow points. Then, oscillatory frequency sweeps were performed at 0.1% strain between 0.1 and 100 rad/s. Frequency sweep measurements were performed at a constant strain in the linear viscoelastic regime (LVR), as determined by the oscillation strain sweep (amplitude sweep) measurement performed for each sample. Within the linear viscoelastic region, in fact, the material response is independent of the magnitude of the deformation and the material structure is maintained intact; this is a necessary condition to accurately determine the mechanical properties of the material.

## Release experiments: in vitro and ex vivo set-up and HPLC drug quantification

Formulations and free drug enema were tested in vitro and ex vivo with vertical diffusion cells (PermeGear, Pennsylvania, USA) using a 3000 nm polycarbonate membrane (Sterlitech Corporation, USA). HEPES buffer (8 mL) with a pH 7.4 or HEPES buffer enriched with 10% (v/v) of EtOH was used as the release medium for TOFA and TAC, respectively and the device was placed in a shaking incubator at 100 rpm and 37 °C. To investigate the effect of lipase on drug release, porcine pancreatic lipase (1000 U/mL,) was added to the sample in the donor chamber. Ex vivo experiments were performed using rat intestinal tissue to evaluate the drug release of our TIF-Gel. Briefly, fresh intestinal tissue was obtained and cut into suitable samples (2 mm*1 mm*1 mm) for Franz cell apparatus. Tissue was placed on the polycarbonate membrane for tensile loading. At designated time points (0.5, 1, 1.5, 2, 3, 4, 5, 6, 7, 8 h), the release medium (HEPES buffer with a pH 7.4 in case of TOFA or HEPES buffer enriched with 30% (v/v) EtOH in case of TAC) was completely replaced with 8 mL of fresh medium, and 1 mL aliquot was taken for lyophilization. Each sample was resuspended with internal standard in the mobile phase and the drug content was analysed by HPLC (details are reported in the SI). The same experimental design was used for both formulations. Moreover, samples of TIF-Gel containing drug were stored at room temperature and 4 °C for 30 days. The drug stability was determined by HPLC analysis. The same experimental design was used for both drugs.

## In vivo investigation

**Chemically induced colitis based on application of DSS.** Female 6–8 week-old C57B6/J mice were ordered from Charles River, Germany and maintained under specific and opportunistic pathogen-free (SOPF) microbiota conditions at the animal facility of the University of Bern (conditions: mice housed in IVC (Techniplast at $20 \pm 2$ °C, humidity $50 \pm 10\%$, 12 h light/12 dark light cycle, each cage is provided with a wooden chew stick, nesting material, and red housing). Females were used to allow randomization of animals within cages (total $n = 45$, group numbers are provided in the various experimental descriptions). Mice were ear-marked, randomly assigned to different cages and treatment groups, and bedding mixed between all cages to avoid potential cage effects on microbiota and to semi-blind the researcher

performing measurements. All methods used were approved by Bernese Cantonal Veterinary Office (permission no. BE 20/18). One day prior to the start of DSS supplementation, mice were intra-rectally injected with 100 µL of empty gel, TOFA in 1% methylcellulose, or gel loaded with drug (5 mg/100 µL).

The next day, drinking water was supplemented with 2% w/v dextran sodium sulfate (DSS; MP Biomedicals, 160110). Every day or every other day, the different compounds were applied intra-rectally until the end of the experiment (see Supplementary Fig. 6). During the experiment, the mice were constantly monitored and the weight and disease scores were recorded when appropriate. Disease score was determined by grading of 1–4 of the following criteria (with grade 4 corresponding to most unhealthy/abnormal): posture, mobility, fur appearance, weight, stool consistency, and stool color, as previously described[74]. At the termination of the experiment, mice were euthanized by asphyxia with carbon dioxide, and organs were collected and used as described in the results. Swiss rolls[75] were made from the colons, fixed overnight with 10% formalin in PBS, then washed with PBS, embedded in paraffin, and sectioned with H&E staining. Histopathology scoring was performed by board-certified pathologist, in a blinded manner, using the following criteria: loss of goblet cells, crypt abscesses, epithelial erosions, hyperemia, thickness of mucosa, and cellular infiltration (maximal score for each criterion: 3)[76].

### Flow cytometry and quantification of single cells
The gating strategy was adapted from previously published work[77]. Briefly, mouse spleens (after weighing) and mesenteric lymph nodes were homogenized through a 70 µm cell strainer, after which point the red blood cells were removed from the spleen by re-suspending the cell pellet in ACK lysing buffer (150 mM $NH_4Cl$, 10 mM $KHCO_3$, 0.1 mM; pH: 7.4) at room temperature for 5 min. Splenocytes were quantified using a CASY cell counter (Omni Life Sciences) and the following populations were quantified following single cell and live/dead selection (ThermoFischer, L34961). T cells (defined as CD3ε + cells; antibody used: eBioscience, 25-0031-82, dilution in PBS: 1:200); dendritic cells (CD11c + , CD11b + ; Biolegend 117324 & 101241, dilutions 1:400, 1:200); neutrophils (CD11b + ; Ly6G + ; Biolegend, B156884, 1:100), macrophages (CD11b + , CD11c-, Ly6G-, Ly6Clo), inflammatory monocytes (CD11b + , Ly6Chi; Biolegend, 128024, 1:100), and eosinophils (CD11b + , CD11c-, Ly6G-, Siglec F + ; BD Bioscience 552126; 1:100). Stained cells were analysed on a BD Bioscience LSR II SORP flow cytometer with FACSdiva software.

### T-cell transfer colitis
To induce T cell-mediated colitis, CD4 + T cells were isolated from the spleen of C57BL/6 mice using the CD4 T cell isolation kit from Stemcell Technologies (# 19852; Cologne, Germany) and subsequently naive T helper cells (CD3 + , CD4 + , CD25low, CD6Lhigh, CD44low cells) were sorted on a FACS Aria III (Becton Dickinson; Eysins, Switzerland) as described previously[54–56]. In all, 12–15 week-old male and female $Rag^{-/-}$ mice in the C57BL/6 background (originally purchased from Taconic from which a local colony was maintained in our vivarium) were injected intraperitoneally with $2.5 \times 10^5$ naive T helper cells. Mice (6 animals/group) were randomly assigned to different cages and treatment groups, and bedding mixed between all cages to avoid potential cage effects on microbiota. All methods used were approved by the animal welfare authority (permission no. ZH043/2021). Starting on day 3 post T cell injection, mice received rectal instillations (100 µL) of empty TIF-Gels, TAC-loaded TIG-Gels or TAC in vehicle solution (1% nitrocellulose in distilled water) once per day until the end of the experiment. Weight development and disease activity scores were measured on a daily basis. On the last day of the experiment (day 18), the mice were anaesthetized using a mixture of ketamine 90–120 mg/kg bodyweight (Vétoquinol, Bern, Switzerland) and xylazine 8 mg/kg

bodyweight (Bayer, Lyssach, Switzerland) and subjected to mouse endoscopy to assess the extent of endoscopic colitis as described previously[63] using the following parameters: (1) thickening of the colon wall, (2) vascularization/bleeding, (3) extent of fibrin deposits, (4) granular appearance of the colon wall, (5) stool consistency. Each parameter was given a score from 0 (normal) to 3 (most severe appearance), resulting in a maximal total score of 15. After colonoscopy, the mice were sacrificed and colon tissue harvested for histology and isolation of *lamina propria* immune cells. Immune cells were isolated from the colon, mesenteric lymph nodes and the spleen, and analysed for immune cell subsets as described previously[78].

### H&E staining and histological analysis of colitis severity
To assess the microscopic extent of colitis, formalin-fixed, paraffin-embedded sections of the most distal 1.5 cm of the colon were subjected to hematoxilin and eosin (H&E) staining using standard protocols[63]. The sections were analysed by two blinded scientists for the extent of epithelial damage (score 0–4) and infiltration of immune cells (score 0–4) resulting in a maximal possible score of 8. Images were taken using a Zeiss Axio Imager.Z2 microscope (Zeiss), equipped with an AxioCam HRc (Zeiss, Jena, Germany) camera and ZEN imaging software (Zeiss, Germany).

### Analysis of cytokine levels in colon
To analyse cytokine levels in the colons, colon pieces were lysed in PBS (1 ml PBS/ 0.1 mg tissue) using a GentleMACS device from Miltenyi Biotec (Miltenyi Biotec, Bergisch Gladbach, Germany). Lysates were then analysed for cytokines using the Bio-Plex Pro Mouse Cytokine 23-plex Assay from Bio-Rad (Hercules, CA) according to the manufacturer's instructions.

### Pharmacokinetics (PK)
The PK studies in healthy animals were performed by the Platform of Biopharmacy of the University of Montreal, in accordance with local animal welfare committee of the University of Montreal, and in agreement with regulations of the Canadian Council on Animal Care (CCAC). Healthy female C57BL/6 mice (5 animals/group) received, under anesthesia, a single intra-rectal administration (100 µL) of either drug-loaded TIF-Gel (TIF-Gel-TOFA or TIF-Gel-TAC) or free drugs (TOFA or TAC in suspension). All the formulations contain 5 mg of TOFA or 1 mg of TAC and they were applied once at $t = 0$ rectally. Plasma levels were determined after 0.25, 1, 2, 4, 6, 12, 24 and 48 h post-application. Animals were euthanized with $CO_2$ after the last sampling point. Blood was collected and stored in K2-EDTA BD-Microtainer™ (Fisher Scientific AG, Switzerland). Drugs were extracted from plasma and their concentration was determined using LC-MS/MS analysis (see Supplementary information). $AUC_{0-48h}$ were calculated according to the trapezoid method.

### Statistics and reproducibility
All statistical analyses performed are available for each experiment in the results section of the manuscript. When possible, animals were randomly assigned to groups and the researchers were blinded (e.g., histology scoring). No data were excluded from analysis. No statistical methods were used to predetermine sample sizes.

### Reporting summary
Further information on research design is available in the Nature Portfolio Reporting Summary linked to this article.

## Data availability
All data generated or analysed during this study are included in this published article and its supplementary information files, or are available from the corresponding authors upon reasonable request. Source data are provided with this paper.

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

## Acknowledgements

Dr. Serena Rosa Alfarano and Ms. Francesca Victorelli (Laboratory of Food & Soft Materials, Institute of Food, Nutrition and Health, IFNH; Department for Health Sciences and Technology, D-HEST, ETH Zurich Switzerland) are kindly acknowledged for their precious support during the SAXS experiments. We thank the team of the Translational Research Unit of the Institute of Tissue Medicine and Pathology. The authors gratefully acknowledge the Innovation Office of the University of Bern for the financial support of the in vivo study. PK is supported by grants from the Swiss National Science Foundation (310030_189185) and by The Bern University Research Foundation. RAG received funding from a "Seal of Excellence Fund" (SELF) from the University of Bern.

## Author contributions

M.C.: investigation, validation and visualization (LMPs preparation, release experiments and rheology), writing-original draft, formal analysis. M.R.S.: investigation, validation and visualization; writing—original draft. R.A.G.: investigation, validation and visualization, writing—original draft, formal analysis. R.M.: writing—review & editing. K.H.: investigation (ex vivo imaging). A.M.: formal analysis. P.K.: conceptualization; supervision; project administration; funding acquisition; writing—review & editing. G.R.: conceptualization; supervision; project administration; funding acquisition; writing—review & editing. P.L.: conceptualization; supervision; project administration; funding acquisition; writing—review & editing. S.A.: conceptualization; data curation; methodology; validation; formal analysis; investigation; writing—original draft; visualization.

## Competing interests

The authors M.C., M.R.S., R.A.G., R.M., K.H., A.M., P.K., P.L. and S.A. declare no competing interests. G.R. declares the following competing interests: consulting to Abbvie, Arena, Augurix, BMS, Boehringer,

Calypso, Celgene, FALK, Ferring, Fisher, Genentech, Gilead, Janssen, Lilly, MSD, Novartis, Pfizer, Phadia, Roche, UCB, Takeda, Tillots, Vifor, Vital Solutions and Zeller; received speaker's honoraria from Abbvie, Astra Zeneca, BMS, Celgene, FALK, Janssen, MSD, Pfizer, Phadia, Takeda, Tillots, UCB, Vifor and Zeller; received educational grants and research grants from Abbvie, Ardeypharm, Augurix, Calypso, FALK, Flamentera, MSD, Novartis, Pfizer, Roche, Takeda, Tillots, UCB and Zeller. Gerhard Rogler is cofounder and head of the scientific advisory board of PharmaBiome. The TIF-Gel technology has been patented. *Patent applicant:* University of Bern and University of Zurich. *Name of inventor(s):* Marianna Carone, Marianne R. Spalinger, Robert A. Gaultney, Philippe Krebs, Gerhard Rogler, Paola Luciani, Simone Aleandri. *Application number:* European Patent Application No 22197842.3 *Status of application:* The above application has been filed with the European Patent Office (EPO). Filing Date: 26.09.2022 *Specific aspect of manuscript covered in patent application:* In vitro, ex vivo and in vivo results.
