## [Peer Review File · Nature Communications]

REVIEWER COMMENTS

Reviewer #1 (Remarks to the Author):

Manuscript title:

Temperature-triggered in situ forming gel for local treatment of ulcerative colitis

Review

The manuscript by Marianna Carone and co-authors reports on drug delivery system for the topical treatment of UC. This platform is based on a temperature-triggered in situ forming lipid gel (TIF-Gel), loaded with tofacitinib or tacrolimus.

Since my specific expertise is on Soft Matter, even if have carefully read the ex-vivo experiments I am not able to provide an expert opinion on this section as I am for the design and structural characterization of the material

Overall, the results show a definite therapeutic improvement, with encouraging results to improve the quality of life of patients.

I believe that this is a nice piece of work, very sound and robust from an experimental point of view. While the structural transition of MLO mesophases from lamellar to cubic structures is already known, its application and the rectal temperature trigger are both elegant and useful for the intended application

This work demonstrates a novel approach which deserves -in my opinion- publication in this journal after some revisions, as below noted

Define better what you mean by “a lower structural strength”, referred to lamellar mesophases in comparison to cubic ones. I believe that this concept should be better quantified by figures and maybe a threshold which defines the ideal formulation. I understand that this is defined by lower value of storage modulus and loss modulus (and I agree), but I guess that some figures should be given.

Figure 1 is very nice yet crowded. For instance, inset C is barely readable. Either remove or enlarge it.

What is the shelf-life of the lamellar phase? I see that the drug stability was checked over 30 days, but I was wondering about the structural stability of the lamellar phase...

Why only the first Bragg reflection of the lamellar phase is detected? This peak is also very broad.

Is the transition reversible if T is brought back to 25? Not that it would matter for application but it would be good to know.

What is the release mechanism of the drug? Dissolution of the depot?

I am no expert, but how well does HEPES reproduce the colon milieu? What is the ionic strength that best matches the biological fluids?

The authors say that the lamellar phase rapidly evolves into the cubic gel once T is raised. How rapid is rapidly?

The abstract is, in my opinion, a bit unattractive. I am not asking for hype, but just to a more specific reference to the novelty of this work.

In addition, I do not like very much the discussion separated from the results, as the manuscript risks resembling a technical report. This might be my personal opinion, but please consider this option.

Reviewer #2 (Remarks to the Author):

The authors developed a gel against ulcerative colitis, which is a common inflammatory disease. Here they used a gel based on monolinolein that was observed to undergo phase transition when heated from room temperature to body temperature. This gel was able to entrap two different types of drugs at high encapsulation efficiencies. The authors then used two different animal models to test their gel. There was little difference between the free drug and the drug-loaded gel when using the inflamed colon model, which the authors assigned to their severe treatment regime. The transfer colitis model in contrast showed that the prepared gel has indeed advantages. Overall, this is an interesting manuscript. I have however two main comments:

- 1) Gel and foam to treat ulcerative colitis are already available. The authors highlighted that their gel is cheaper/less toxic/readily available, but there was little discussion on how their gel compared in performance to already reported gels
- 2) I liked the fundamental study using SAXS, but the authors did not really discuss/test how the gels behave in the actual biological environment. What is the phase transition and drug release like when exposed to biological fluid? How much does the gel adhere to the colon and how easily is it detached?

Other comments

Figure 1: There is little discussion on the relationship between chemical structure and gel formation. The authors used monolinolein (MLO), but it is not clear why (although there is some explanation in the discussion) and if this surfactant is unique in this behaviour

Phase transition analysis was measured by SAXS using buffer solutions in the presence of lipase. Are these models valid and would they represent the actual environment, which might be rich in proteins? Moreover, a 84% MLO and 16% water formulation was chosen, but wouldn't the actual composition quickly change in the body when exposed to the milieu.

How is TOFA and TAC present in the gel? Is it a mixture of do they form drug crystals (or drug precipitates)?

I think the drug release needs to be discussed in relation to the thickness of the gel membrane. What dimension does the gel shown in Fig 2 have? I suppose the samples dimension was the same in both cases?

Figure 2: It would have been interesting to see how the drug release rate changes when the release is carried out at different temperatures, below and above the phase transition

Figure 3: What is the statistical significance between TIF-Gel-TOFA and TOFA?

Animal tests: Have the authors measured how quickly the gel is excreted from the mouse?

Reviewer #3 (Remarks to the Author):

This is a well written report of preclinical efficacy of novel compounds of established anti inflammatory agents in models of human IBD.

What's new is the applied materials engineering in a gel compound that harbors two drugs (tacrolimus and tofacitinib) in such a fashion as to result in a depo delivery intraluminal system upon rectal temperature ascertainment.

There is substantial data on the physical properties of the gel, the loading of drug, and the dispersement of drug in a temperature sensitive fashion.

Two preclinical models are selected to demonstrate in vivo applications.

Application of tacrolimus or tofacitinib to DSS and/or T cell dependent models of colitis is not novel (Pharm 2020; 105:541-549; PMID 30668755; 32808031); yet the mode of application is new.

The preclinical data is very modest. In particular the histology pictures of the DSS colitis show substantial inflammation in the gel treated animals and a clear difference between vehicle and tofa treated patients is not clear. It is not clear whether the histology index was in a blinded fashion. The results of the Rb hi transfer system is also quite atypical. Achieving weight loss and clinical inflammation within days and sacrifice of animals at 20 days is quite a bit earlier than nearly all the published literature using this model (typically 6 weeks). This finding would require a bit more discussion; however the effects of drug in this system is not robust.

Of greater interest to the community would be the pharmacokinetics/pharmacodynamics of gel loaded drug. A major advance in the field would be less systemic toxicity when using potent drugs such as tacro

and Jak inhibitors. Systemic levels of tacro have been reported with rectal administration (PMID 24809233); thus this avenue would be a very interesting and important contribution.

Point-by-point reply to reviewers' comments:

For your convenience, point-by point changes are detailed in the following responses (**blue font**) to the reviewers' comments and/or questions (**bold font**). We also included page references to the main text, which is cited here in the responses using quotation marks and italic, with changes from the original submission additionally in red font.

Reviewer #1 (Remarks to the Author):

The manuscript by Marianna Carone and co-authors reports on drug delivery system for the topical treatment of UC. This platform is based on a temperature-triggered in situ forming lipid gel (TIF-Gel), loaded with tofacitinib or tacrolimus. Since my specific expertise is on Soft Matter, even if have carefully read the ex-vivo experiments I am not able to provide an expert opinion on this section as I am for the design and structural characterization of the material. Overall, the results show a definite therapeutic improvement, with encouraging results to improve the quality of life of patients. I believe that this is a nice piece of work, very sound and robust from an experimental point of view. While the structural transition of MLO mesophases from lamellar to cubic structures is already known, its application and the rectal temperature trigger are both elegant and useful for the intended application. This work demonstrates a novel approach which deserves -in my opinion- publication in this journal after some revisions, as below noted.

We thank the reviewer for the positive evaluation of our work.

Define better what you mean by “a lower structural strength”, referred to lamellar mesophases in comparison to cubic ones. I believe that this concept should be better quantified by figures and maybe a threshold which defines the ideal formulation. I understand that this is defined by lower value of storage modulus and loss modulus (and I agree), but I guess that some figures should be given.

We thank the reviewer for this comment. We agree that the meaning of “a lower structural strength” could profit from different phrasing and additional data. Following the reviewer's suggestions, we have revised the text and Figure 1. Specifically, we added a **new panel “e”** to Figure 1 showing the flow and yield points obtained for the lamellar and cubic gels by amplitude sweep experiments. Both parameters better define the differences between our low viscous lamellar precursor and the high viscoelastic cubic gel obtained after rectal administration and they might quantify a threshold above which a formulation would be too viscous to be rectally applied.

Changes to the manuscript, page 8:

Moreover, either the flow or the yield points (both representing the shear limit above which a material starts to behave like a fluid) better determine the differences between our low viscous lamellar precursor and the high viscoelastic cubic gel and they might identify a threshold above which a formulation is too elastic to be rectally applied (Fig 1e). Since the sliding of a lamellae can occur along any possible direction, a low shear is required to be applied to this gel so that it starts behaving like a fluid and it can be forced to pass easily through a canula for enema, a syringe or a colon pipe. This translates into a low viscosity material with a low structural strength easier to administer compared to the fully hydrated cubic gel owing to its high flow and a yield point (see also SI; Figure S2).

The new figure and the revised caption are reported below for convenience.

Fig. 1. In vitro characterizations of the TIF-Gel: a) schematic depiction of the in vitro characterization and the mechanism of the gel formation. b) SAXS spectra acquired at different temperatures (25, 30 and 38 °C; bottom, middle and top spectra, respectively) on gels containing increasing amount of water (12%, 14%, 16% and 18 % w/w) and (c) SAXS spectra acquired at different times (5, 10, 20 and 30 min) after incubation at 38 °C; (d) frequency sweep at the end (purple symbols) and beginning (grey symbols) of the release experiments; (e) Flow and yield points obtained for lamellar phase (grey bars) and for cubic gel (purple bars); (f) SAXS before (1) and after incubation of LMPs in HEPES buffer (2) and in HEPES buffer enriched with 1000 U/mL of lipase (3). The LMPs cartoons (L; cubic ia3d, cubic pn3m and hexagonal) are adapted from Aleandri et al.³⁸.

Figure 1 is very nice yet crowded. For instance, inset C is barely readable. Either remove or enlarge it.

We thank the reviewer for this comment and in agreement with their suggestion we have revised Figure 1. The old panel c (partial phase diagram) was removed and as requested, a new panel e now shows the flow and yield point obtained for lamellar and cubic gel. These parameters, extrapolated from amplitude sweep experiments, can help the reader better understand the concept of “a lower structural strength” as noted above. Moreover, to help readability of the figure, we simplified the color code: purple, green, and grey represent measurements carried out at 38 °C, 30 °C and 25 °C, respectively.

What is the shelf-life of the lamellar phase? I see that the drug stability was checked over 30 days, but I was wondering about the structural stability of the lamellar phase.

We thank the reviewer for this comment. We carried out a stability study of the lamellar phase after 30 days. As shown in the new Figure S10, the lamellar geometry does not change after 30 days of storage at room temperature.

Fig. S10. In vitro characterizations of the TIF-Gel: SAXS spectra acquired at 25 °C immediately after the preparation (T₀) and after 30 days (T_{30 days}). The calculate lattice parameter at T₀ and T_{30 Days} are 4.8 and 4.6 nm, respectively. The slight decrease in the measured lattice parameter can be explained by a loss of minimal amount of water from the sample which was stored in a 2 mL microcentrifuge conical tube.

This information was added to the text (page 25) and to the SI:

Noteworthy, MLO is relatively inexpensive and available in large quantities at a high grade of purity (Good Manufacturing Practice or Food Grade); the manufacturing is a single-step process, and both the tested drugs and the TIF-Gel exhibit long-term stability (see SI, Figure S9 and S10).

Why only the first Bragg reflection of the lamellar phase is detected? This peak is also very broad.

We thank the reviewer for this comment. The reason why only the first Bragg reflection was observed in the case of lamellar structures is linked to the sensitivity of the instrument and to the acquisition time used. Indeed, the second and the third reflections (at 3.5 and 4.5 nm⁻¹) of the lamellar phase are not as pronounced as the first one and they are not as easy to detect. However, an increased acquisition time up to 60 min allowed us to at least observe the second reflection. Consequently, all the SAXS spectra of the lamellar phase in the manuscript (Figures 1 and 2) and the SAXS experimental section were updated. Notably, the lattice parameter calculated using only the first reflection and the one calculated using both reflections do not differ (4.7 ± 1.1 and 4.4 ± 1.6 nm⁻¹, respectively). We have also changed the x-axis scale in Figures 1 and 2 and now the peaks look less broad or at least comparable with other published spectra. In the case of TAC-loaded gel, the first peak appears at higher q compared to those obtained with the TOFA-loaded gel or with the empty gel and this may be due to a shrinking of the lamella induced by the hydrophobic drug.

Is the transition reversible if T is brought back to 25? Not that it would matter for application but it would be good to know.

We appreciate the comment; this input, that we believe to be valuable, should be added to the manuscript. The transition is reversible (new Figure S1) and, as reported in the manuscript, while the rectal temperature (38 °C) transforms the lamellar into a cubic phase ($Ia3d$), cooling down the sample till the room temperature reverts this transition. However, this transition (Q→L) is not as fast as the first and ca. 30 min are needed for the sample to revert to the lamellar phase.

Fig. S1. In vitro characterizations of the TIF-Gel: SAXS spectra acquired at different temperatures: at 25 °C (bottom), after 30 min equilibration at 38 °C (middle) and after 30 min equilibration at 25 °C (top).

This aspect was clarified in the manuscript and this set of data added in the Supplementary Information (Figure S1).

Changes in the manuscript at page 8:

The transition is reversible if the temperature is brought back to 25 °C (see SI; Figure S1). While this information is not relevant for rectal applications *per se*, it is an important property for the storage conditions of TIF-Gel.

What is the release mechanism of the drug? Dissolution of the depot?

We did not observe any gel erosion (no weight loss was recorded during the *in vitro* and *ex vivo* experiments) and we can, thus, exclude that the release process is driven by gel dissolution. We are prone to believe that the release is regulated by the log P and the molecular weight of the drug; by the symmetry of the mesophase; and by the ability of the drug to pass through the lipidic layers and diffuse out in the release media.

We added a paragraph which explains this aspect (pages 10- 11):

In 2015, Martiel et. al. developed the structural control efficiency index (SCEI)⁵⁶, which provides an estimate of the kinetics of drug release for various phases. However, in our case, the phase identity of the gel changes dynamically during the release experiment. Thus, we cannot use the above-mentioned paradigm to describe the release profile. Indeed, our hydrophobic drugs do not follow a Fickian diffusion profile and, consequently, the release profile cannot be modeled using the Higuchi equation. We did not observe any gel erosion (no weight loss was recorded either *in vitro* or *ex vivo* experiments) and we can, therefore, reject the hypothesis that the release process is driven by gel dissolution.

I am no expert, but how well does HEPES reproduce the colon milieu? What is the ionic strength that best matches the biological fluids?

We thank the reviewer for this relevant question. While different release media simulating the small intestine (and the ascendent colon section) are reported and even commercially available (such as the fasted states simulated intestinal fluids; FaSSIF and fed state simulated intestinal fluids; FeSSIF), less is known about the characteristics of the biological fluids in the colorectal tract. Further, no commercially

available simulated rectal medium exists. To bypass this limitation, lipase (an esterase enzyme secreted in case of inflammation) was used in the release experiment to enrich the media and to closely mimic an inflamed colon milieu, as recently described (*Science Translational Medicine*, 2015, 7:300ra128) to test the drug release from a thermo-responsive gel during ulcerative colitis. We also employed an *ex vivo* approach in which tissues isolated from healthy rat colon were used as natural membrane and the gel was kept in contact with the luminal side of the colon during the release test.

In the revised version of the manuscript the results of a new set of *in vivo/ex vivo* experiments were included to fill the gaps between the results obtained *in vitro* (using a buffer) and the real environment of the colon. Overall, the phase transition of the gel (see SI; Figure S3), its adhesion to the colon wall (see SI; Figure S8), and the pharmacokinetic of the loaded drugs (new Figure 6) are now evaluated after a rectal application of the gel *in vivo*.

The authors say that the lamellar phase rapidly evolves into the cubic gel once T is raised. How rapid is rapidly?

We appreciate the question since one of the strengths of our TIF-gel is indeed its rapid transformation from a lamellar (L) phase into a cubic phase (Q). As it can be seen from Figure 1c, the reflection characteristics of this L phase adopt those characteristic of a Q phase **after only 5 min** of incubation at 38 °C, indicating a fast conversion of the lamellar precursor into the Ia3d cubic structure (page 7). Yet, we do not know if the gel is already formed after an even shorter incubation time. Indeed, we need at least 5 min to acquire a SAXS spectrum where the Ia3d reflections are visible, and during this time the sample is incubated at 38 °C. We can also assume that the jellification process can be even faster *in vivo*, as the gel can use both the rectal temperature and the available water as triggers to form the high viscous cubic gel *in situ*.

The fast L→Q conversion is a key point of our formulation, and we further highlight this crucial property at the beginning of the result section and in Figure1 panel a in the updated manuscript.

Changes in the manuscript at page 6 and Figure 1a.

Once applied into the rectum, the precursor L phase gradually absorbs heat (and the available amount of water) from the body and rapidly (<5 min) converts into the cubic phase, contributing to the formation of a depot *in situ*. The gel, therefore, allows local release of the incorporated drug in a sustained fashion.

The abstract is, in my opinion, a bit unattractive. I am not asking for hype, but just to a more specific reference to the novelty of this work.

We thank the reviewer for the suggestion. We have revised the abstract accordingly.

In addition, I do not like very much the discussion separated from the results, as the manuscript risks resembling a technical report. This might be my personal opinion, but please consider this option.

We thank the reviewer for this comment, and we also considered merging the results and discussion as suggested during the preparation of the manuscript. However, in this specific case, we believe that the

reader might benefit from the chosen format, since what we would like to deliver, more than the description of the results and their discussions, is the applicability of the TIF-gel as a valuable platform to treat UC. Thus, although we agree that the manuscript risks resembling a technical report, we prefer to not change the manuscript's structure. Moreover, looking at *Nature Communications* papers, it seems that, a majority of the studies published therein, discussion is separated from the results.

Reviewer #2 (Remarks to the Author):

The authors developed a gel against ulcerative colitis, which is a common inflammatory disease. Here they used a gel based on monolinolein that was observed to undergo phase transition when heated from room temperature to body temperature. This gel was able to entrap two different types of drugs at high encapsulation efficiencies. The authors then used two different animal models to test their gel. There was little difference between the free drug and the drug-loaded gel when using the inflamed colon model, which the authors assigned to their severe treatment regime. The transfer colitis model in contrast showed that the prepared gel has indeed advantages. Overall, this is an interesting manuscript.

We thank the reviewer for the positive evaluation of our work.

I have however two main comments:

1) Gel and foam to treat ulcerative colitis are already available. The authors highlighted that their gel is cheaper/less toxic/readily available, but there was little discussion on how their gel compared in performance to already reported gels.

We thank the reviewer for this suggestion. We agree that a direct comparison between the TIF-Gel and its competitors is pivotal and for this reason we already reported in the manuscript a comprehensive description of all the pros of our gel.

For convenience we list below all the advantages of our TIF-Gel with respect to the available foam, enema, and gels already in the manuscript (here in black), together with the new sentences added to the manuscript (here in red):

- We demonstrated that our newly developed TIF-Gel can host and release drugs of different polarity in a sustained manner, and the rectal temperature can be employed as stimulus to transform the plastic (and low viscosity) lamellar precursor into the structured (and high viscoelastic) cubic gel in situ, **while no available enema or foam can control the drug release (page 23).**
- On the other hand, the pseudoplastic precursor has a higher viscosity than commercially available enemas such as Asacol® and Pentasa® and foam-containing 5-ASA and budesonide. **Thus, once applied, our TIF-Gel adheres to the colon wall and it is retained for 6 h into the intestine, a time needed to avoid loss of material^{39,65,66} (see SI; Figure S8) (page 23).**
- In comparison to the available FluidCrystal technology® developed by the Swedish company Camurus^{67,68}, which consists of an alcoholic lipid solution that transforms to a gel upon contact

with water, TIF-gel is not only dependent on water content, but also uses temperature as a trigger for an in situ gelation. This aspect is of particular relevance since the volume of rectal fluid is low and highly affected by age, biological sex and pathology. In addition, TIF-gel is less fluid than a lipidic solution thereby avoiding a loss of material upon rectal application and, in contrast with liquid crystal technology, no initial burst release of drug was observed with our gel. Indeed, from our observations regarding drug retention, we are prone to believe that the release is regulated by the log P and molecular weight of the drug; and by the ability of the drug to pass across the lipidic layers and diffuse into the environment (page 24).

- Compared with other depot systems, our gel presents several potential advantages. First, TIF-Gel is made from a nontoxic, generally recognized as safe (GRAS) compound and, differently from other in situ forming implants (made by triglycerol monostearate or poly(lactic-co-glycolic acid), PLGA) currently in clinical trials or commercially available, it does not require organic solvents (that may be toxic) during its in situ formation. Differently from what observed with other in situ forming gels, the L→Q transition is not associated with an initial burst release of the drug (page 25).
- A low volume (0.1 mL) of TIF-Gel is able to deliver high amounts of drugs (up to 10 mg). Differently from the recently developed hydrogel for local drug delivery in IBD in which a low proportion of dexamethasone was encapsulated⁷¹, our bulk gel is able to provide 100% drug loading (5 mg TOFA/100 mg gel and 1 mg TAC/100 mg gel) and still ensure a sustained release in vitro, ex vivo and in vivo (page 24).
- Moreover, 10% w/w of both TOFA and TAC can be loaded without affecting the phase identity and the transition temperature of the gel, which gives a lamellar phase at room temperature and a cubic (Ia3d) phase at 38 °C. These drug concentrations are higher than those contained in the commercially available enemas for UC treatment, which contain only 4% w/w of 5-ASA (in case of Asacol[®]) or 2% w/w of budesonide (for Budenofalk[®] and Entocort[®]). Although the drug concentrations in the TIF-Gel can reach high levels, this delivery platform also helps to simultaneously minimize systemic circulation (page 26).

2) I liked the fundamental study using SAXS, but the authors did not really discuss/test how the gels behave in the actual biological environment. What is the phase transition and drug release like when exposed to biological fluid?

We thank the reviewer for this question. While different release media simulating the small intestine are reported, and even commercially available (such as the fasted states simulated intestinal fluids FaSSIF and fed state simulated intestinal fluids FeSSIF), less is known about the biological fluids of the colorectal tract and, to date, no rectal simulated fluid has been marketed. For the descending colon only animal models are available and are mostly used in pre-clinical studies.

To bypass this limitation, as described in the methods of the original manuscript, we employed an *ex vivo* approach in which tissues isolated from healthy rat colon were used as natural membrane. The gel was kept in contact with the luminal side of the colon during the release test. To more closely mimic the

inflamed colon milieu, lipases (enzymes secreted in correspondence of inflammation events) were used in the release experiment, as recently described in a study reporting about a thermoresponsive gel for ulcerative colitis (Science Translational Medicine, 2015, 7:300ra128).

Predicting the phase transition of the gel when exposed to biological fluid only relying on the *in vitro* experiments is challenging, yet we believe that the reviewer raised a very relevant matter. To answer their question, we performed an additional *in vivo* experiment in which healthy animals were administered 100 μ L of TIF-Gel and either the excreted gel (with stool) or the residual gel present into the colon after 6 h was collected and analyzed by SAXS (animal were sacrificed, the colon harvested, and the residual gel washed 3x with PBS before analysis). As shown in the new Figure S3, the Bragg reflections characteristic of L phase were present before administration at 25 °C, whereas the gel excreted with the stool (collected 30 min after the rectal application) showed the L \rightarrow la3d transition. Moreover, the lamellar phase absorbed heat and water during the experiment reaching a cubic (pn3m) phase, as already observed in the *in vitro* investigations (see Figure 1, panel f).

Figure S3. *In vivo* characterizations of the TIF-Gel: SAXS spectra acquired at different time points (before administration, excreted with the stool and the residual gel present into the colon) at 38 °C.

Changes in the manuscript at page 9:

These transitions were also confirmed *in vivo* where, after rectal application, the gel excreted and collected with the stool after 30 min had an la3d phase identity, whereas the residual gel present in the colon after 6 h was determined to be in the pn3m cubic phase (see SI; Figure S3).

Moreover, in the revised version of the manuscript, a new set of *in vivo* experiments were performed to evaluate the drugs release and their absorption into the blood stream in which tofacitinib, in particular, is released into the circulation less readily when loaded in the TIF-Gel (see pharmacokinetics data; new Figure 6).

How much does the gel adhere to the colon and how easily is it detached?

We performed further *in vivo/ex vivo* experiments to answer this important question from the reviewer.

As shown in the new **Figure S8**, once applied intrarectally, the viscous gel is concentrated towards the distal colon after 2 h, with an accompanying trend towards signal reduction after 6 h, as determined by quantification of the radiant efficiency of each tissue sample. Fluorescence above untreated background

was detectable in all treated tissues at 6 h post injection, thus indicating that the gel adheres to the colon for at least this time period.

Figure S8. The TIF-Gel adheres to healthy colonic tissue for at least 6 hours. a) Experimental scheme depicting the procedure: Healthy animals received an enema of 100 μ L of DiR-TIF-Gel. Animals were sacrificed after 30 min, 2 and 6 h and the colon was harvested and imaged (b). c) The obtained signal was analysed as radiant efficiency (RE), which was normalized to radiant efficacy recorded at 30 min.

The description of the study shared above was added to the main manuscript (page 23), together with additional information as detailed below:

On the other hand, the pseudoplastic precursor has a higher viscosity than commercially available enemas such as Asacol[®] and Pentasa[®] and foam-containing 5-ASA and budesonide. Thus, once applied, our TIF-Gel adheres to the colon wall and it is retained for at least 6 hours, a time needed to avoid loss of material^{39,66,67} (see SI; Figure S8).

Figure 1: There is little discussion on the relationship between chemical structure and gel formation. The authors used monolinolein (MLO), but it is not clear why (although there is some explanation in the discussion) and if this surfactant is unique in this behaviour.

We appreciate this comment, and we have answered it by providing more details about the relationship between chemical structure of monolinolein and gel formation in the revised manuscript. The critical packing parameter (CPP; defined as the ratio between the hydrophobic chain volume and the hydrophobic chain length in the molten state times the cross-sectional molecular area) can qualitatively predict the structural phase given by each lipid in the presence of water. However, because of the complexity of lipidic mesophases structures, CPP theory cannot fully describe the system. Lipids such as phytantriol (PHT), monoolein (MO), and monolinolein (MLO) are the most well-known macromolecules capable of forming lipidic mesophases in water via self-assembly. MO and MLO, which are generally recognized as safe for human and animal use by the US Food and Drug Administration, are the most widely used materials for encapsulating a broad range of drugs with various sizes and polarities (*Physics Today* 73, 7, 38. 2020.). Among these lipids, the phase diagrams of MLO (for reviewers' eyes only: **Figure i**, panel c) are unique; at any temperature and water content there is no coexistence of different mesophases. Consequently, changing the temperature (without incrementing the water content) results in a transformation from lamellar to cubic (ia3d) structure. On the contrary, for neither monoolein nor phytantriol the rectal temperature can be used as stimulus to obtain the lamellar → cubic transition.

Figure i. The phase diagrams of the lipids (a) monoolein, (b) phytantriol, and (c) monolinolein. The light orange areas in the phase diagrams show the coexistence of two (or more) phases. (Adapted from Aleandri and Mezzenga, *Physics Today* 73, 7, 38. 2020.)

A clear explanation of the reason why MLO was selected is now present in the manuscript.

Changes in the manuscript, page 23:

Among the various acyl glycerol lipids (such as phytantriol and monoolein) capable of forming lipidic mesophases in water via self-assembly, MLO (affirmed as GRAS for human and animal use by the US Food and Drug Administration) is a widely used material for encapsulating a broad range of drugs with various sizes and polarities. Its phase diagrams are unique, as at any temperature and water content it does not present coexistence of mesophases⁶⁵. Consequently, changing the temperature (without incrementing the water content) results in a direct transformation from lamellar to ia3d cubic structure.

Phase transition analysis was measured by SAXS using buffer solutions in the presence of lipase. Are these models valid and would they represent the actual environment, which might be rich in proteins?

We thank the reviewer for this valuable comment. We are aware that, *in vivo*, different enzymes, proteins and bacteria (such as intestinal microbiota) might interact with our gel. From this rich and complex milieu, lipase is commonly used in the *in vitro* release experiment to more closely mimic the inflamed colon environment and it was recently employed to test the drug release from a thermoresponsive gel for ulcerative colitis (*Science Translational Medicine*.12 August 2015 Vol 7 Issue 300 300ra128).

Each *in vitro* approach is characterized by specific limitations, though. We performed an additional *in vivo* experiment (as already mentioned above) in which healthy mice were administered with 100 μ L of TIF-Gel and either the excreted gel (with stool) or the residual gel present into the colon at the end point (after 6 h the animal was sacrificed, and colon harvested) was collected and analyzed with SAXS. As shown above (see new Figure S3) the Bragg reflections characteristic of L phase were present before administration at 25 °C, whereas the gel excreted with the stool (collected 30 min after the rectal application) shown a Ia3d cubic phase. The lamellar phase absorbed heat and water during the experiment reaching a cubic (pn3m) phase as was previously observed in the *in vitro* investigations (see Figure 1 panel f).

Moreover, a 84% MLO and 16% water formulation was chosen, but wouldn't the actual composition quickly change in the body when exposed to the milieu.

We selected this specific composition for different reasons. Firstly, a gel composed by 16% water gives a lamellar to cubic transition exactly at rectal temperature. Secondly, the rheological properties of this gel composition (84% lipid/ 16% water) are optimal for rectal application and, in comparison to the available FluidCrystal technology[®] developed by the Swedish company Camurus (which consists of an alcoholic lipid solution that transforms to a gel upon contact with water), our lamellar gel is a more viscous material which avoids loss of material upon rectal application.

As correctly pointed out by the reviewer, once applied, the gel will also absorb the available amount of water (which is low and highly affected by age, biological sex, and pathology) and transform into the structured cubic gel. The phase change will not be only dependent on water content, but it will also employ temperature as a trigger for an *in situ* jellification. This translates to a faster L→Q transition (less than 5 min) than that of a less hydrated gel (water content <16%).

How is TOFA and TAC present in the gel? Is it a mixture of do they form drug crystals (or drug precipitates)?

We thank the reviewer for this comment. Both drugs are dissolved into the gel matrix and do not form crystals once accommodated into the lipidic gel (at least at the amount used in this study) as proven by the absence of reflections associated with a drug crystallization in the WAXS (wide angle X-ray scattering) spectra (new Figure S4, panel a). We carried out additional experiments to assess whether both drugs were homogeneously distributed into the gel matrix. To determine this, the gel (loaded with

TAC or TOFA) was prepared as described in the manuscript and transferred into a 2 mL microcentrifuge conical tube. The tube was centrifuged and kept at rest for 24 h. After that, the gel was divided in 3 different layers (Top, Middle and Bottom) and the drug content of each layer quantified. As shown in the new Figure S4 (panel b), we did not detect any difference in drug amounts between the different layers, confirming that TOFA and TAC were homogeneously distributed through the gel.

Figure S4. Drugs distribution into TIF-Gel. a) WAXS spectra obtained for empty gel (bottom), TOFA loaded gel (middle) and TAC loaded-gel (top). All the WAXS spectra (acquired for 30 min at 25 °C) show a broad shoulder (and no evident peak) which indicates the amorphous state of the lipid chain and absence of crystalline structures. b) Homogeneity of drugs into the gel. The amount of drug present in 3 different gel layer (Top, Middle and Bottom) was quantified by HPLC.

This aspect was clarified in the manuscript (page 10) and in the SI (page S4):

Moreover, both drugs do not form crystals once embedded in the 3D gel structure as proven by the absence of reflections associated with a drug crystallization in the WAXS spectra. The drugs were homogeneously distributed into the gel matrix (see SI; Figure S4).

I think the drug release needs to be discussed in relation to the thickness of the gel membrane. What dimension does the gel shown in Fig 2 have? I suppose the samples dimension was the same in both cases?

While the release processes of hydrophilic molecules (a Fickian diffusion process follows a first-order kinetic profile) depend on the structure of the mesophase and on the thickness of the gel membrane, in the case of hydrophobic drugs it is not easy to model the release profile. As reported above in our reply to reviewer #1, in 2015 researchers introduced the structural control efficiency index (SCEI; Martiel et al., J. Control. Release 204, 78. 2015) which provides an estimate of the kinetics of drug release for various phases having different symmetries and permeability to drugs.

In our case, however, since our phase dynamically changes during the release experiment, we cannot use the above-mentioned theories. We believe that the release is regulated by the log P and the molecular weight of the drug, by the symmetry of the mesophase and by the ability of the drug to pass across the lipidic layers and diffuse out in the release media.

The reviewer correctly noticed that we did not discuss the drug release in relation to the thickness of the gel membrane. The reason for this is linked to the technical limitations of our release setup. During our release studies we always placed the same amount of sample (100 μ L) in the donor chamber of a Franz cell apparatus using an insulin syringe. Because of this setup, it was not possible to keep the thickness of the mesophase constant through the different experiments and thus it was not possible to calculate the diffusion coefficients. Moreover, in the real-life *in vivo* rectal application of the gel (which will be applied similarly to an enema) there will be no control on the thickness layer of the TIF-gel adhering to the colon wall.

Overall, considering the complexity of the gel, its administration, and the experimental limitations, we decided to not speculate on the role played by the thickness of the gel membrane in the release. Nevertheless, we decided to add a brief explanation about the release process to the manuscript.

Changes in the manuscript at pages 10-11:

In 2015, Martiel et. al. developed the structural control efficiency index (SCEI)⁵⁶, which provides an estimate of the kinetics of drug release for various phases. However, in our case, the phase identity of the gel changes dynamically during the release experiment. Thus, we cannot use the above-mentioned paradigm to describe the release profile. Indeed, our hydrophobic drugs do not follow a Fickian diffusion profile and, consequently, the release profile cannot be modeled using the Higuchi equation. We did not observe any gel erosion (no weight loss was recorded either *in vitro* or *in ex vivo* experiments) and we can, therefore, reject the hypothesis that the release process is driven by gel dissolution.

Figure 2: It would have been interesting to see how the drug release rate changes when the release is carried out at different temperatures, below and above the phase transition.

We agree with the reviewer that this would be an interesting point to evaluate. The release experiments were carried out in presence of release media (2 mL of buffer surround the gel in the donor chamber of the vertical Franz cell apparatus). Therefore, even if the temperature is kept below the L \rightarrow Q transition, the lamellar phase would still absorb water with time and transform itself into the cubic gel, as shown in phase diagram in Figure i (the lamellar phase does not exist in excess of water). We expect therefore the release rate to be affected by the water intake during the experiment and the consequent phase change.

Figure 3: What is the statistical significance between TIF-Gel-TOFA and TOFA?

As treatment with TOFA alone led to a disease with a severity between the TIF-gel and TIF-gel TOFA mice, the numbers of animals here did not provide enough power to distinguish between those groups in all of the tested variables; in addition to the fact that testing three groups simultaneously diminishes statistical robustness. Yet, there are several experimental parameters of disease measurement that are different between empty gel and drugged-gel mice ($p < 0.05$) that do not meet that same threshold for empty-gel vs. free drug mice; for example, disease score, colon length, and spleen weights. When $p < 0.1$ and > 0.05 , we have now included those values in Figure 3 comparing TOFA and TIF-Gel-TOFA to underline this trend. In addition, we have performed cytokine analysis and thereby identified several

relevant inflammatory mediators that merit inclusion in the manuscript (see new panel e in Figure 3). Here, we were able to detect cytokine differences not only between the empty-gel-treated mice, but also between the free drug and gel-drug groups. These data, combined with the new data shown in Figure 6, strongly suggest a longitudinal suppression of inflammation at the local site of gel application, which in turn has an effect on systemic parameters (e.g. spleen weight), and that histology analyses towards the end became less different, also since this type of analysis is semi-quantitative.

Animal tests: Have the authors measured how quickly the gel is excreted from the mouse?

We thank the reviewer for this question. This parameter was evaluated in a preliminary *in vivo* study during which the mouse was kept in a recovery cage (without any bedding) for 30 min after rectal application (100 μ L of empty gel) and then moved back in the experimental cage. The gel excreted with the stool (within this half an hour) was collected and weighted. As reported in Figure ii (for reviewers' eyes only), we could estimate approximately half of the administered gel was defecated after ca. 10 min. It is important to remark that, after administration, the mice were left free to move and this introduced a certain external variability in the excretion time of the stools and/or of the gel. These mice also were non-colitic and, thus, had more feces present in the colon that could force out some gel. Mice and/or patients with more advanced colitis may have reduced material in the colon exert this force. Before patient treatment, the ideal proportion of applied material would have to be estimated via pilot studies in human colitis patients.

Figure ii. Amount of gel excreted with feces after 10 min. After rectal application of 100 μ L of the gel, healthy mice were kept into an observing cage and the amount of gel excreted with face collected and analyzed.

Reviewer #3 (Remarks to the Author):

This is a well written report of preclinical efficacy of novel compounds of established anti-inflammatory agents in models of human IBD. What's new is the applied materials engineering in a gel compound that harbours two drugs (tacrolimus and tofacitinib) in such a fashion as to result in a depo delivery intraluminal system upon rectal temperature ascertainment. There is substantial data on the physical properties of the gel, the loading of drug, and the dispersement of drug in a temperature sensitive fashion. Two preclinical models are selected to demonstrate

in vivo applications. Application of tacrolimus or tofacitinib to DSS and/or T cell dependent models of colitis is not novel (Pharm 2020; 105:541-549; PMID 30668755; 32808031); yet the mode of application is new.

We thank the reviewer for the positive evaluation of our work.

The preclinical data is very modest. In particular the histology pictures of the DSS colitis show substantial inflammation in the gel treated animals and a clear difference between vehicle and tofa treated patients is not clear. It is not clear whether the histology index was in a blinded fashion.

To address this reviewer's concern, we performed an additional cytokine measurement of snap-frozen colon tissue samples taken from this experiment. We were able to identify robust differences in several clinically relevant pro- and anti-inflammatory cytokines (Figure 3, new panel e), further supporting the applicability of our drug-gel mixture. We also saw differences in mouse disease activity scores at days 5 and 6, but these had disappeared by the end of the experiment, which may explain why the histology index values were all high during analysis. It should also be noted that spleen weight (a marker of runaway inflammation) was significantly reduced in the treated mice (40% reduction). This is likely not due to systemic effects of the drug, but rather due to a secondary effect of the reduced local inflammation, as our release data (Figure 6) show a low amount of TOFA leakage from the intestine. The histology analysis was indeed performed by a blinded board-certified pathologist, as detailed in our methods. We have also clarified this in the figure legend.

Changes in the manuscript at pages 15-16:

Furthermore, local pro-inflammatory cytokine levels were reduced in TOFA and TIF-Gel-TOFA-treated mice, and anti-inflammatory IL-10 levels were increased only in the TIF-Gel-TOFA group (Fig. 3e).

Changes in the manuscript at page 25:

The gel efficacy is further supported by the changes in cytokines measured from colon tissue samples, in which the most robust changes in these inflammatory mediators were observed in the TIF-Gel-TOFA treated mice.

The new Figure 3 and the revised caption are reported below for convenience.

Fig. 3. TIF-Gel-TOFA effectively mitigates intestinal inflammation and disease induced by DSS treatment in mice. Mice were prophylactically treated rectally with either empty gel (TIF-Gel), tofacitinib in vehicle (TOFA), or TOFA loaded-gel (TIF-Gel-TOFA) and thereafter challenged with 2% DSS in the drinking water. Treatments were then applied every other day until the end of the experiment. Weights (a) and disease score (b) were recorded throughout the experiment. At the end of the experiment, spleens, mesenteric lymph nodes (mLNs) and colons were removed from the mice. The spleens were weighed (c) and single splenocytes were enumerated (d). The tissue concentrations of various cytokines were measured (e). The mouse colon length was measured (f), and the colon was opened transversally,

cleaned, and prepared for histology (g). Colon histopathology scores were determined by a blinded pathologist and aggregated (H). *: $p < 0.05$, **: $p < 0.01$, ***: $p < 0.001$, ****: $p < 0.0001$, and actual value is provided for values less than 0.1 but not meeting significance threshold as determined by 2-way ANOVA (a), multiple Student's- tests with Holm-Sidak correction for multiple comparisons (b), and one way ANOVA with multiple comparisons and Tukey correction (c, d, e, f, h). All tests were performed using Prism (GraphPad) and applying default settings for the above-mentioned analyses; naïve values were excluded from analyses; all error bars are \pm SEM.

The results of the Rb hi transfer system is also quite atypical. Achieving weight loss and clinical inflammation within days and sacrifice of animals at 20 days is quite a bit earlier than nearly all the published literature using this model (typically 6 weeks). This finding would require a bit more discussion; however, the effects of drug in this system is not robust.

We thank the reviewer for this comment. As in all animal models of colitis, the disease severity and kinetics of disease development in the T cell transfer model of colitis (here we defined naïve CD4+ T cells as CD62Llow/CD44high and did not sort based on Rb expression) is highly dependent on environmental factors and especially the microbiota composition crucially shapes the kinetics and severity of the disease. In our animal facility, observation of first signs of disease 10-14 days post injection of naïve T cells is typical. Nevertheless, colitis development was still rather modest at the time of sacrifice of the animals in this particular experiment (weight loss around 5%, disease activity scores between 2-4, endoscopic colitis scores of around 10, and only modestly enlarged crypts and modest infiltration visible in H&E stained sections of the colon), thus a longer experimental duration might have resulted in clearer disease development, however prolonging the experiment was not possible due to animal welfare legislation. Regarding the comment about the lack of robustness of the effects of the drug in this system, we wish to point out that it is obvious that rectal application of tacrolimus alone had minor effects on weight development and disease scores, however, there were robust differences in endoscopic and histological scores as well as colon length. Furthermore, both administration forms of tacrolimus resulted in clear effects on the immune cell composition, indicating that there is a clear and robust effect of the drug in our disease setting, although the effect was greater when the drug was administered using the TIF-gel. Moreover, as above, we performed an additional cytokine measurement of snap-frozen colon tissue samples taken from this experiment. We found reduced levels of IFN- γ and IL-17 in mice treated with the free drug in colonic tissues (Figure 5d). TIF-Gel-TAC further reduced levels of these two cytokines and in addition also significantly reduced levels of TNF- α , indicating that TIF-Gel-TAC has a stronger effect on the production of inflammatory cytokines when compared to the free drug.

Changes in the manuscript at pages 18-19:

These findings were also reflected in cytokine measurements in colonic tissues (Figure 5d), where we found reduced levels of IFN- γ and IL-17 in mice treated with the free drug. TIF-Gel-TAC further reduced levels of these two cytokines and in addition also significantly reduced levels of TNF- α (Figure 5d),

indicating that TIF-Gel-TAC has a stronger effect on the production of inflammatory cytokines when compared to the free drug.

The new Figure 5 and the revised caption are reported below for convenience.

Fig. 5. Immune cell populations and cytokine levels in the colon from TAC-loaded TIF-Gel (TIF-Gel-TAC) treated mice. 12–15-week-old Rag^{-/-} mice develop colitis via transfer of 2.5 x10⁵ naïve T cells. Starting on day 3 post T cell transfer, mice received daily rectal instillations with TIF-Gel without drug (TIF-Gel), TAC-loaded TIF-Gels (TIF-Gel-TAC) or TAC in vehicle (TAC). Depicted is the relative abundance of the indicated cell populations in (a) the colonic lamina propria, (b) mesenteric lymph nodes; and (c) the spleen on day 19 after T cell transfer, and (d) levels of indicated cytokines in colon

lysates. * $p < 0.05$, ** $p < 0.01$, *** $p < 0.001$ as determined by one way ANOVA with multiple comparisons and Tukey correction. All tests were performed using Prism (GraphPad) and applying default settings for the above-mentioned analyses; all error bars are \pm SEM.

Of greater interest to the community would be the pharmacokinetics/pharmacodynamics of gel loaded drug. A major advance in the field would be less systemic toxicity when using potent drugs such as tacro and Jak inhibitors. Systemic levels of tacro have been reported with rectal administration (PMID 24809233); thus this avenue would be a very interesting and important contribution.

We thank the reviewer for this comment, and we also believe that a pharmacokinetic (PK) study is an important aspect to further assess the potential clinical suitability of our TIF-Gel and to demonstrate that there is indeed less systemic drug absorption (and thus less toxicity) when the drug is loaded in a gel. Accordingly, the study was conducted as requested by this Reviewer. Our new Figure 6 shows the systemic concentrations of the two drugs when applied with or without the gel, with low levels detected in both cases, and a statistically significant reduction in systemic TOFA levels when the TIF-gel was employed.

Changes in the manuscript at pages 21-22:

Rectal drug delivery via TIF-Gel reduces systemic drug exposure

To demonstrate that rectal TIF-Gel application was indeed suitable to minimizing systemic exposure, we analyzed drug release in vivo by longitudinally monitoring in mice drug plasma levels after TIF-Gel application into the colon. For this purpose, naïve received a single enema of either drug-loaded TIF-Gel (TIF-Gel-TOFA or TIF-Gel-TAC) or of free drugs (TOFA or TAC) and plasma drug concentrations were measured at different time points (Fig. 6a). Mice receiving free TOFA had an early peak in plasma concentration at 0.25 h (Fig. 6b); TOFA plasma levels rapidly decreased thereafter, following first-order kinetics. In mice receiving TIF-Gel-TOFA, the peak concentration at 0.25 h was significantly lower. The area under the curve (AUC), a measurement of cumulative systemic drug absorption, was also significantly reduced in the mice treated with TIF-Gel-TOFA when compared to the group treated with free TOFA (Fig. 6d). Administration of TAC, either as free drug, or as drug loaded gel resulted in a low (and negligible) systemic drug circulation (Fig. 6c) and no difference was detected in their AUCs (Fig. 6e).

Changes in the manuscript at pages 26-27:

Although the drug concentrations in the TIF-Gel can reach high levels, this delivery platform also helps to simultaneously minimize systemic circulation. Indeed, as demonstrated by our pharmacokinetics study, the plasma concentration of the TOFA in TIF-Gel-treated mice was lower than in animals rectally treated with free drug within a 48 h period after application. On the other end, the pharmacokinetics of TAC either as free drug or as drug-loaded gel results in a negligible systemic drug absorption and no difference was detected in their AUCs. In a previously reported clinical study, rectal administration of

TAC via a suppository resulted in a systemic exposure with a relative bioavailability of ~70% after 24 h in comparison to the oral formulation⁶³. Here, upon administration, patients were asked to remain in a semi-recumbent position for 3 h (without defecation), a practice that facilitates drug absorption. Differently from this study, after administration, our mice were left free to move and this introduced a certain external variability in the excretion time of the stools and/or of the drug. The low residence time in the intestine could explain the low plasmatic levels of free TAC observed, whereas the low drug absorption of TAC released from TIF-Gel loaded could be explained by a slow and continuous drug release from the gel. In conclusion, this reduced systemic drug concentration indicates that the use of TIF-Gel could reduce the number and severity of systemic side effects resulting from the use of potent drugs such as TOFA and TAC in a clinical setting.

The new Figure 6 is reported below for convenience.

Fig. 6. Drug delivery via TIF-Gel leads to a low systemic drug exposure. (a) Experimental design for the pharmacokinetic study. Naïve mice (n=5/group) received a single enema of either drug-loaded TIF-Gel (TIF-Gel-TOFA or TIF-Gel-TAC) or free drugs (TOFA or TAC). The plasma drug concentrations were measured at the indicated time points after administration. Plasma concentration versus time profiles of the pharmacokinetic experiment of TOFA- (b) and TAC-treated animals (c) and Area Under

the Curve (AUC)_{0-48h} values of TOFA- and TAC-treated mice (d and e, respectively). ***p<0.001, as determined by Student's t test.

REVIEWERS' COMMENTS

Reviewer #1 (Remarks to the Author):

I believe that the authors have addressed the issues raised by the reviewers.

In my opinion, the submission has been consistently improved and is now suited for publication in this journal.

Reviewer #2 (Remarks to the Author):

This is a very well executed revision. The authors provided additional experiments and made changes to the manuscript. I think the work can be published as is

Reviewer #3 (Remarks to the Author):

The authors should be congratulated for a concise and well constructed response to prior reviews. No further concerns.